# IEBins: Iterative Elastic Bins for Monocular Depth Estimation

**Shuwei Shao[1], Zhongcai Pei[1], Xingming Wu[1], Zhong Liu[1], Weihai Chen[2,*] and Zhengguo Li[3]**

[1]School of Automation Science and Electrical Engineering, Beihang University, China
[2]School of Electrical Engineering and Automation, Anhui University, China
[3]SRO department, Institute for Infocomm Research, A*STAR, Singapore
*Corresponding author, email:whchen@buaa.edu.cn

## Abstract

Monocular depth estimation (MDE) is a fundamental topic of geometric computer vision and a core technique for many downstream applications. Recently, several methods reframe the MDE as a *classification-regression* problem where a linear combination of probabilistic distribution and bin centers is used to predict depth. In this paper, we propose a novel concept of **iterative elastic bins (IEBins)** for the classification-regression-based MDE. The proposed IEBins aims to search for high-quality depth by progressively optimizing the search range, which involves multiple stages and each stage performs a finer-grained depth search in the target bin on top of its previous stage. To alleviate the possible error accumulation during the iterative process, we utilize a novel elastic target bin to replace the original target bin, the width of which is adjusted elastically based on the depth uncertainty. Furthermore, we develop a dedicated framework composed of a feature extractor and an iterative optimizer that has powerful temporal context modeling capabilities benefiting from the GRU-based architecture. Extensive experiments on the KITTI, NYU-Depth-v2 and SUN RGB-D datasets demonstrate that the proposed method surpasses prior state-of-the-art competitors. The source code is publicly available at `https://github.com/ShuweiShao/IEBins`.

## 1 Introduction

Monocular depth estimation (MDE) is a long-standing and fundamental topic in geometric computer vision, with many applications in robotics [1], 3D reconstruction [2], scene understanding [3], etc. It consists in inferring the depth map from a single RGB image, which is ill-posed and has the challenge of scale ambiguity, because the same 2D image can be projected from infinitely many 3D scenes. Recently, more and more learning-based approaches have been proposed to promote the development of MDE [4, 5, 6, 7, 8, 9, 10, 11]. Without loss of integrity, these methods can be grouped into three categories: *regression*, *classification* and *classification-regression* [12].

**Regression** is the most primitive and straightforward formulation [4, 7, 13, 14, 15, 16, 8], which directly generates continuous pixel-wise depth under the supervision of a regression loss. Despite its great success as a universal paradigm, the regression-based model suffers from unsatisfactory results [5]. **Classification** is proposed in [5] and [17] to formulate the MDE as per-pixel classification and predict the optimal depth interval. More specifically, it discretizes the full depth range into multiple intervals (bins) in uniform/Log-uniform space (Fig. 1 (a) and (b)) and takes the bin center (depth candidate) of classified target bin as the final depth prediction. While the classification makes the MDE easier and significantly improves the model performance, the poor visual quality with discontinuity artifacts tends to appear [6].

37th Conference on Neural Information Processing Systems (NeurIPS 2023).

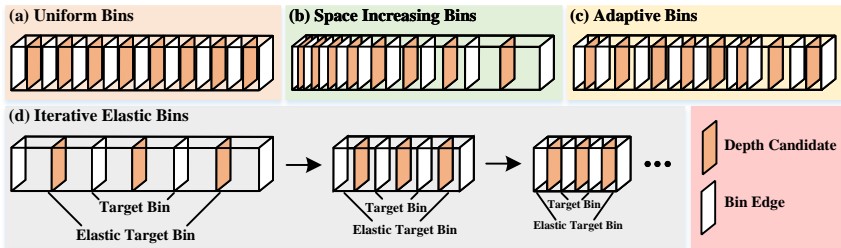

Figure 1: Illustration of different bin types comprising uniform bins [5], space increasing bins [5], adaptive bins [6] and the proposed iterative elastic bins.

In order to overcome the discontinuity artifacts in classification, some methods [6, 18, 12, 19, 20] reframe the MDE as a per-pixel **classification-regression** problem, learning the probabilistic distribution on each pixel and using the linear combination with depth candidates as the final depth prediction. Theoretically, it can achieve the sub-pixel depth estimation. On top of that, Bhat *et al.* [6] noticed the extreme fluctuations in depth distribution across different scenes and proposed to derive adaptive bins (Fig. 1 (c)) from the image content. Li *et al.* [12] and Bhat *et al.* [19] further improved [6] by disentangling bins generation and probabilistic distribution learning or performing local predictions of depth distributions in a gradual step-wise manner. Moreover, Agarwal *et al.* [20] developed a bin center predictor that uses pixel queries at the coarsest level to predict bins.

In this paper, we introduce a novel concept termed **iterative elastic bins (IEBins, Fig. 1 (d))**, tailored for the classification-regression-based MDE. The IEBins leverages multiple small number of bins, instead of one standard number of bins, to search for high-quality depth by progressively reducing the search range. To specify, the proposed IEBins involves multiple stages, where each stage predicts depth at different granularities and performs a finer-grained depth search in the target bin (the bin in which the predicted depth is located in our case) of its previous stage. Unfortunately, the depth ground-truth may fall outside the target bin due to wrong depth predictions, resulting in unstable optimization and degraded accuracy. To mitigate the error accumulation during iterations, we elastically adjust the width of the target bin according to the depth uncertainty that indicates the potential depth errors. Inspired by [21], we utilize the variance of the probabilistic distribution to quantify the uncertainty. Last but not least, we develop a dedicated framework (Fig. 2) consisting of two main components: a feature extractor that generates strong feature representations and a gated recurrent unit (GRU)-based iterative optimizer with powerful temporal context modeling capabilities that predicts the per-pixel probabilistic distribution from its hidden state for the depth classification-regression.

To summarize, our main contributions are three-fold:

- We introduce a novel iterative elastic bins strategy for the classification-regression-based MDE. The IEBins performs an iterative elastic search using multiple small number of bins in light of the depth uncertainty.
- We develop a framework to instantiate the proposed IEBins, where a feature extractor attains strong feature representations and a GRU-based iterative optimizer predicts the probabilistic distribution.
- Extensive experiments are conducted on the KITTI [22], NYU-Depth-v2 [23] and SUN RGB-D [24] datasets, and the experimental results show that the proposed method exceeds previous state-of-the-art competitors.

## 2 Related work

**Monocular depth estimation.** Learning-based MDE has witnessed tremendous progress in recent years. Saxena *et al.* [27] proposed a pioneering work that uses Markov Random Field to capture critical local- and global-image features for depth estimation. Later, Eigen *et al.* [4] introduced a convolutional neural network (CNN)-based architecture to attain multi-scale depth predictions. Since then, CNNs have been extensively studied in MDE. For instance, Laina *et al.* [28] utilized residual CNN [29] for better optimization. Recently, Transformer [30] has attracted widespread attention in the computer vision community [31, 32, 33]. Following the success of visual Transformer in

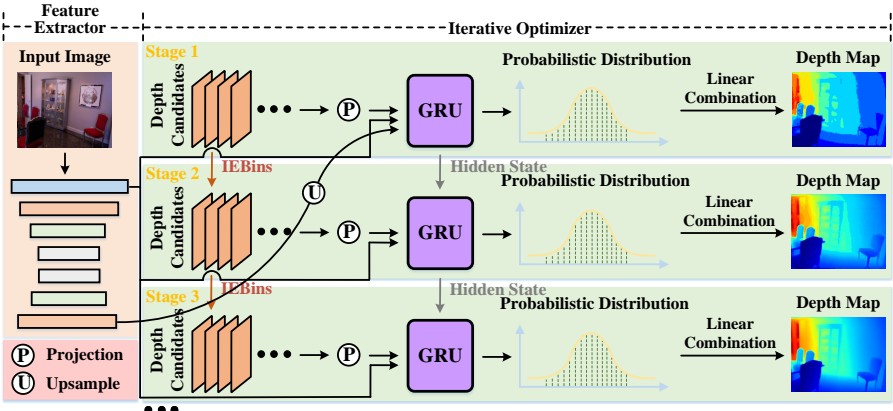

Figure 2: An overview of the whole framework. The upsample stands for the pixel shuffle [25]. The projection is achieved using four $3 \times 3$ convolutional layers followed by the ReLU activation [26].

other tasks, Yang *et al.* [14], Ranftl *et al.* [34] and Yuan *et al.* [8] replaced CNN with Transformer, further improving the performance. However, the above methods suffer from sub-optimal solutions induced by the inherent drawback of regression [5]. Fu *et al.* [5] and Cao *et al.* [17] proposed to formulate MDE as a classification problem and discretized the full depth range into multiple bins to predict the optimal depth interval. Diaz *et al.* [35] softened the classification label in [5] during the training phase. In addition, Bhat *et al.* [6], Johnston *et al.* [18], Li *et al.* [12], Bhat *et al.* [19] and Agarwal *et al.* [20] reframed the MDE as per-pixel classification-regression to mitigate the discontinuity artifacts caused by depth discretization. Among them, [6] introduced an adaptive bins strategy to boost the performance. In contrast, we propose an iterative elastic bins paradigm for the classification-regression-based MDE, which uses multiple small number of bins rather than one standard number of bins like [6]. [19] also refines the binning structure in an iterative manner. However, [19] divides all bins from the previous stage at each stage, and when the stage increases, the number of bins increases, while the proposed IEBins locates and divides the target bin only, and the number of bins is not changed at different stages.

**Iterative refinement.** Recently, Teed *et al.* [36] proposed to iteratively refine a displacement vector field through the GRU for optical flow estimation, emulating the first-order optimization. This idea gradually receives attention in other tasks, such as stereo [37, 38], structure from motion [39] and scene flow [40]. In this paper, we deploy a GRU-based iterative optimizer to predict the per-pixel probabilistic distribution, where the hidden state is updated at each stage for more accurate estimation.

## 3  Methodology

In this section, we first introduce the iterative elastic bins tailored for the classification-regression-based MDE. Then, we demonstrate a detailed description regarding the feature extractor and iterative optimizer in our framework. Finally, we present the training loss function.

### 3.1  Iterative Elastic Bins

The proposed IEBins embodies the idea of iterative division of bins, and is composed of two parts, **initialization** and **update**. In the initialization stage, we perform a coarse and uniform discretization of the full depth range. During each subsequent stage, we follow an iterative process to locate and uniformly discretize the target bin by using the target bin as the new depth range. The details are presented below.

**Initialization.** To initialize the IEBins, we discretize the full depth range $[d_{\min}, d_{\max}]$ into $N$ bins in the uniform space,

$$e_n = d_{\min} + nB, n = 0, 1, ..., N, \tag{1}$$

with

$$B = \frac{d_{\max} - d_{\min}}{N}, \tag{2}$$

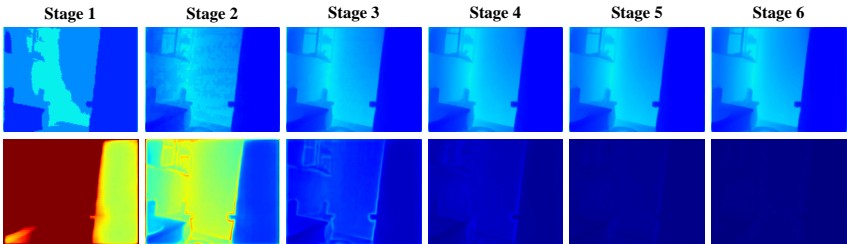

| Stage 1 | Stage 2 | Stage 3 | Stage 4 | Stage 5 | Stage 6 |

Figure 3: Illustration of depth and uncertainty maps for each stage. The upper row is depth maps, which we display using the bin center of every target bin to better visualize the refinement process. The lower row is uncertainty maps (yellow/red: high/highest uncertainty; blue: low uncertainty).

where $e_n$ denotes the $n$-th bin edge, $B$ denotes the bin width, $d_{\max}$ and $d_{\min}$ are set to 80 and 0.1, and 10 and 0.1 for KITTI [22] and NYU-Depth-v2 [23] datasets, respectively, and $N$ is set to 16 when not otherwise specified, a significantly fewer number of bins than the 256 bins used in AdaBins [6]. As we cannot directly use discrete bins to predict depth, we take the bin centers to represent the depth candidates of bins,

$$\mathcal{D}_n = \frac{e_n + e_{n+1}}{2}, n = 0, 1, ..., N - 1, \tag{3}$$

where $\mathcal{D}_n$ denotes the $n$-th depth candidate. Once the per-pixel probabilistic distribution associated with depth candidates is predicted, we can acquire a depth prediction via their linear combination

$$\widehat{\mathbf{D}}(\mathbf{p}) = \sum_{n=0}^{N-1} \mathcal{D}_n \cdot \mathbf{P}_n(\mathbf{p}), \tag{4}$$

where $\widehat{\mathbf{D}}(\mathbf{p})$ denotes the predicted depth, $\mathbf{p}$ denotes the pixel coordinate and $\mathbf{P}_n$ denotes the $n$-th depth probability.

**Update.** Each subsequent stage searches at a finer-grained granularity in the target bin of its previous stage. The target bin represents the corresponding bin in which the predicted depth is located, which we obtain by comparing the predicted depth with bin edges

$$e_n \leq \widehat{\mathbf{D}}(\mathbf{p}) < e_{n+1} \tag{5}$$

and denote as $[e_n, e_{n+1}]$. However, as discussed in the introduction section, the depth ground-truth may fall outside the target bin due to depth prediction errors. In this case, the errors will gradually accumulate as the stage increases, resulting in unstable optimization and decreased accuracy. To account for such a failure case and make the paradigm more robust, we propose to leverage the elastic target bin to update the depth candidates. The width of the elastic target bin is adjusted flexibly based on the depth uncertainty reflecting the likelihood of depth inaccuracies. Inspired by [21], we capture the uncertainty via the variance of the probabilistic distribution. In particular, the variance $\widehat{\mathbf{V}}(\mathbf{p})$ is calculated as

$$\widehat{\mathbf{V}}(\mathbf{p}) = \sum_{n=0}^{N-1} \left( \mathcal{D}_n - \widehat{\mathbf{D}}(\mathbf{p}) \right)^2 \cdot \mathbf{P}_n(\mathbf{p}). \tag{6}$$

The corresponding standard deviation is $\widehat{\sigma}(\mathbf{p}) = \sqrt{\widehat{\mathbf{V}}(\mathbf{p})}$. Given the target bin $[e_n, e_{n+1}]$ and standard deviation $\widehat{\sigma}(\mathbf{p})$, we can acquire the elastic target bin by

$$[e_n - \kappa\widehat{\sigma}(\mathbf{p}), e_{n+1} + \kappa\widehat{\sigma}(\mathbf{p})] \tag{7}$$

where $\kappa$ is a coefficient that determines the error tolerance and is set to 0.5. For a pixel with severe depth errors, the standard deviation will become larger such that the elastic target bin has higher immunity against errors. On top of that, we renew $d_{\min}$ and $d_{\max}$ as $e_n - 0.5\widehat{\sigma}(\mathbf{p})$ and $e_{n+1} + 0.5\widehat{\sigma}(\mathbf{p})$, respectively. Through Eqs.1, 2, 3 and 4, we can achieve the updated depth candidates and prediction. The update step will be repeated until reaching to the final stage. It should be noted that although we do not show $\mathbf{p}$ behind $\mathcal{D}$ in above formulas, the depth candidates $\mathcal{D}$ can actually be spatially-varying due to different elastic target bins in subsequent update steps. In Fig. 3, we display examples of depth maps and associated uncertainty maps at each stage.

## 3.2 Feature Extractor

The feature extractor adopts an encoder-decoder structure with skip-connections. The encoder uses the recently proposed Swin-Transformer family backbones [33, 41], which take an RGB image of size $H \times W$ as input and generate a four-level feature pyramid. Then, the skip-connections propagate the pyramid features into the decoding phase. The decoder uses three neural conditional random field (CRF) modules [8] to capture the vital long-range correlation. We alternate between the neural CRF module and pixel shuffle [25] up to the $\frac{H}{4} \times \frac{W}{4}$ resolution. Next, the $\frac{H}{4} \times \frac{W}{4}$ resolution feature maps from the encoder and the decoder are respectively sent to the iterative optimizer as the context feature and the initialization of GRU hidden state.

## 3.3 Iterative Optimizer

To efficiently predict the probabilistic distribution at each stage, we deploy a GRU-based iterative optimizer taking inspiration from [36], since the GRU is capable of retaining the information from history stages and can fully exploit the temporal context during iterations. The optimizer operates at $\frac{H}{4} \times \frac{W}{4}$ resolution.

More specifically, we first project the depth candidates $\mathcal{D}$ into the feature space using four $3 \times 3$ convolutional layers and each convolutional layer is followed by a ReLU activation [26]. We then concatenate the projected feature and the context feature to constitute a tensor $\mathbf{I}^k$ as the input. The structure inside GRU is

$$\mathbf{z}^{k+1} = \text{sigmoid}\left(Conv_{5\times 5}\left(\left[\mathbf{h}^k, \mathbf{I}^k\right], W_z\right)\right), \tag{8}$$

$$\mathbf{r}^{k+1} = \text{sigmoid}\left(Conv_{5\times 5}\left(\left[\mathbf{h}^k, \mathbf{I}^k\right], W_r\right)\right), \tag{9}$$

$$\widehat{\mathbf{h}}^{k+1} = \tanh\left(Conv_{5\times 5}\left(\left[\mathbf{r}^{k+1} \odot \mathbf{h}^k, \mathbf{I}^k\right], W_h\right)\right), \tag{10}$$

$$\mathbf{h}^{k+1} = \left(1 - \mathbf{z}^{k+1}\right) \odot \mathbf{h}^k + \mathbf{z}^{k+1} \odot \widehat{\mathbf{h}}^{k+1}, \tag{11}$$

where $k$ is the stage index, $\mathbf{z}$ is the update gate, $\mathbf{r}$ is the reset gate, $Conv_{5\times 5}$ is the separable $5 \times 5$ convolution, $\odot$ is the element-wise multiplication, $W_z$, $W_r$ and $W_h$ stand for learnable parameters. The hidden state $\mathbf{h}$ is initialized by the $\frac{H}{4} \times \frac{W}{4}$ resolution output in the decoder. The probabilistic distribution is finally predicted from the updated hidden state with two $3 \times 3$ convolutional layers. Meanwhile, the feature maps are regularized by ReLU and Softmax activations, respectively. A linear combination of probabilistic distribution and depth candidates is applied to obtain the depth prediction, which is further upsampled to the original resolution by the bilinear interpolation.

Equipped with this optimizer, initiating at a coarse prediction, the predicted depth is iteratively refined and eventually converges to the final result.

## 3.4 Training Loss Function

**Pixel-wise depth loss**. Following [8, 6], we leverage a scaled Scale-Invariant loss for depth supervision [7],

$$\mathcal{L}_{pixel} = \sum_{k=1}^{K} \alpha \sqrt{\frac{1}{|\mathbf{T}|} \sum_{\mathbf{p}} \left(\mathbf{g}\left(\mathbf{p}\right)\right)^2 - \frac{\beta}{|\mathbf{T}|^2} \left(\sum_{\mathbf{p}} \mathbf{g}\left(\mathbf{p}\right)\right)^2}, \tag{12}$$

where $\mathbf{g}\left(\mathbf{p}\right) = \log \widehat{\mathbf{D}}\left(\mathbf{p}\right) - \log \mathbf{D}^{gt}\left(\mathbf{p}\right)$, $K$ is the maximum number of stages and is set to 6, $\mathbf{T}$ stands for the set of pixels having valid ground-truth values, $\alpha$ and $\beta$ are set to 10 and 0.85 based on [7].

# 4 Experiment

We evaluate the proposed method on both outdoor and indoor datasets, which include KITTI [22], NYU-Depth-v2 [23] and SUN RGB-D [24]. In the following, we start by introducing the relevant datasets, evaluation metrics and implementation details. Then, we present quantitative and qualitative comparisons to prior state-of-the-art competitors, generalization and ablation studies, model parameters and inference time comparison.

| Method | Backbone | Abs Rel ↓ | Sq Rel ↓ | RMSE ↓ | RMSE log ↓ | $\delta < 1.25$ ↑ | $\delta < 1.25^2$ ↑ | $\delta < 1.25^3$ ↑ |
|---|---|---|---|---|---|---|---|---|
| DORN [5] | ResNet-101 | 0.072 | 0.307 | 2.727 | 0.120 | 0.932 | 0.984 | 0.994 |
| VNL [42] | ResNeXt-101 | 0.072 | - | 3.258 | 0.117 | 0.938 | 0.990 | 0.998 |
| BTS [7] | DenseNet-161 | 0.060 | 0.249 | 2.798 | 0.096 | 0.955 | 0.993 | 0.998 |
| PWA [43] | ResNeXt-101 | 0.060 | 0.221 | 2.604 | 0.093 | 0.958 | 0.994 | **0.999** |
| TransDepth [14] | R-50+ViT-B/16† | 0.064 | 0.252 | 2.755 | 0.098 | 0.956 | 0.994 | **0.999** |
| AdaBins [6] | E-B5+mini-ViT | 0.058 | 0.190 | 2.360 | 0.088 | 0.964 | 0.995 | **0.999** |
| P3Depth [44] | ResNet-101 | 0.071 | 0.270 | 2.842 | 0.103 | 0.953 | 0.993 | 0.998 |
| NeWCRFs [8] | Swin-Large† | 0.052 | 0.155 | 2.129 | 0.079 | 0.974 | 0.997 | **0.999** |
| BinsFormer [12] | Swin-Tiny | 0.058 | 0.183 | 2.286 | 0.088 | 0.968 | 0.995 | **0.999** |
| BinsFormer [12] | Swin-Large† | 0.052 | 0.151 | 2.098 | 0.079 | 0.974 | 0.997 | **0.999** |
| PixelFormer [20] | Swin-Large† | 0.051 | 0.149 | 2.081 | 0.077 | 0.976 | 0.997 | **0.999** |
| **Ours** | Swin-Tiny | 0.056 | 0.169 | 2.205 | 0.084 | 0.970 | 0.996 | **0.999** |
| **Ours** | Swin-Large† | **0.050** | **0.142** | **2.011** | **0.075** | **0.978** | **0.998** | **0.999** |

Table 1: **Quantitative depth comparison on the Eigen split of KITTI dataset**. We provide results of the proposed method based on Swin-Large and Swin-Tiny backbones [33]. The maximum depth is capped at 80m. R-50 and E-B5 are the abbreviations of ResNet-50 [29] and EfficientNet-B5 [45], respectively. † indicates that the models are pre-trained by ImageNet-22K. '-' means not applicable. The best results are marked in **bold**.

| Method | dataset | SILog ↓ | Abs Rel | Sq Rel ↓ | iRMSE ↓ | RMSE ↓ | $\delta < 1.25$ ↑ | $\delta < 1.25^2$ ↑ | $\delta < 1.25^3$ ↑ |
|---|---|---|---|---|---|---|---|---|---|
| DORN [5] | validation | 12.22 | 11.78 | 3.03 | 11.68 | 3.80 | 0.913 | 0.985 | 0.995 |
| BTS [7] | validation | 10.67 | 7.51 | 1.59 | 8.10 | 3.37 | 0.938 | 0.987 | 0.996 |
| BA-Full [46] | validation | 10.64 | 8.25 | 1.81 | 8.47 | 3.30 | 0.938 | 0.988 | 0.997 |
| NeWCRFs [8] | validation | 8.31 | 5.54 | 0.89 | 6.34 | 2.55 | 0.968 | 0.995 | 0.998 |
| **Ours** | validation | **7.58** | **5.10** | **0.75** | **5.90** | **2.37** | **0.974** | **0.996** | **0.999** |
| DORN [5] | online test | 11.77 | 8.78 | 2.23 | 12.98 | - | - | - | - |
| BTS [7] | online test | 11.67 | 9.04 | 2.21 | 12.23 | - | - | - | - |
| BA-Full [46] | online test | 11.61 | 9.38 | 2.29 | 12.23 | - | - | - | - |
| PWA [43] | online test | 11.45 | 9.05 | 2.30 | 12.32 | - | - | - | - |
| Vip-Deeplab [47] | online test | 10.80 | 8.94 | 2.19 | 11.77 | - | - | - | - |
| NeWCRFs [8] | online test | 10.39 | 8.37 | 1.83 | 11.03 | - | - | - | - |
| PixelFormer [20] | online test | 10.28 | 8.16 | 1.82 | 10.84 | - | - | - | - |
| BinsFormer [12] | online test | 10.14 | 8.23 | 1.69 | 10.90 | - | - | - | - |
| **Ours** | online test | **9.63** | **7.82** | **1.60** | **10.68** | - | - | - | - |

Table 2: **Quantitative depth comparison on the official split of KITTI dataset**. The SILog is the main ranking metric.

## 4.1 Datasets and Evaluation Metrics

**KITTI** is an outdoor dataset captured by equipment mounted on a moving vehicle, providing stereo images and corresponding 3D laser scans. The images are around $376 \times 1241$ resolution. Here we adopt two data splits, Eigen training/testing split [4] and official benchmark split [48]. The former uses 23488 left view images for training and 697 images for testing. The latter consists of 85898 training images, 1000 validation images and 500 test images without the depth ground-truth. The evaluation results on the official benchmark split are generated by the online server.

**NYU-Depth-v2** is an indoor dataset that has RGB images and ground-truth depth maps at a $480 \times 640$ resolution. We evaluate the proposed method on the official data split, which involves 36253 images for training and 654 images for testing.

**SUN RGB-D** is collected from indoor scenes with high diversity using four sensors, containing roughly 10K images. The dataset is only used for zero-shot generalization study and the official 5050 test images are adopted.

**Evaluation metrics.** Similar to previous work [8], we leverage the standard evaluation protocol to validate the efficacy of the proposed method in experiments, i.e., relative absolute error (Abs Rel), relative squared error (Sq Rel), root mean squared error (RMSE), root mean squared logarithmic error (RMSE log), inverse root mean squared error (iRMSE), $\log_{10}$ error ($\log_{10}$), threshold accuracy ($\delta < 1.25$, $\delta < 1.25^2$ and $\delta < 1.25^3$) and square root of the scale invariant logarithmic error (SILog).

| Method | Backbone | Abs Rel ↓ | Sq Rel ↓ | RMSE ↓ | $\log_{10}$ ↓ | $\delta < 1.25$ ↑ | $\delta < 1.25^2$ ↑ | $\delta < 1.25^3$ ↑ |
|---|---|---|---|---|---|---|---|---|
| DORN [5] | ResNet-101 | 0.115 | - | 0.509 | 0.051 | 0.828 | 0.965 | 0.992 |
| VNL [42] | ResNeXt-101 | 0.108 | - | 0.416 | 0.048 | 0.875 | 0.976 | 0.994 |
| BTS [7] | DenseNet-161 | 0.110 | 0.066 | 0.392 | 0.047 | 0.885 | 0.978 | 0.994 |
| PWA [43] | DenseNet-161 | 0.105 | - | 0.374 | 0.045 | 0.892 | 0.985 | 0.997 |
| Long *et al.* [15] | HRNet-48 | 0.101 | - | 0.377 | 0.044 | 0.890 | 0.982 | 0.996 |
| TransDepth [14] | R-50+ViT-B/16† | 0.106 | - | 0.365 | 0.045 | 0.900 | 0.983 | 0.996 |
| AdaBins [6] | E-B5+mini-ViT | 0.103 | - | 0.364 | 0.044 | 0.903 | 0.984 | 0.997 |
| P3Depth [44] | ResNet-101 | 0.104 | - | 0.356 | 0.043 | 0.898 | 0.981 | 0.996 |
| LocalBins [19] | E-B5 | 0.099 | - | 0.357 | 0.042 | 0.907 | 0.987 | **0.998** |
| NeWCRFs [8] | Swin-Large† | 0.095 | 0.045 | 0.334 | 0.041 | 0.922 | **0.992** | 0.998 |
| BinsFormer‡ [12] | Swin-Tiny | 0.113 | - | 0.379 | 0.047 | 0.890 | 0.983 | 0.996 |
| BinsFormer‡ [12] | Swin-Large† | 0.094 | - | 0.330 | 0.040 | 0.925 | 0.989 | 0.997 |
| PixelFormer [20] | Swin-Large† | 0.090 | - | 0.322 | 0.039 | 0.929 | 0.991 | **0.998** |
| **Ours** | Swin-Tiny | 0.108 | 0.061 | 0.375 | 0.046 | 0.893 | 0.984 | 0.996 |
| **Ours** | Swin-Large† | **0.087** | **0.040** | **0.314** | **0.038** | **0.936** | **0.992** | **0.998** |

Table 3: **Quantitative depth comparison on the NYU-Depth-v2 dataset**. The maximum depth is capped at 10m. ‡ stands for that the model is trained using auxiliary scene class information.

| Method | Backbone | Abs Rel ↓ | RMSE ↓ | $\log_{10}$ ↓ | $\delta < 1.25$ ↑ | $\delta < 1.25^2$ ↑ | $\delta < 1.25^3$ ↑ |
|---|---|---|---|---|---|---|---|
| Chen *et al.* [49] | SENet | 0.166 | 0.494 | 0.071 | 0.757 | 0.943 | 0.984 |
| VNL [42] | ResNeXt-101 | 0.183 | 0.541 | 0.082 | 0.696 | 0.912 | 0.973 |
| BTS [7] | DenseNet-161 | 0.172 | 0.515 | 0.075 | 0.740 | 0.933 | 0.980 |
| AdaBins [6] | E-B5+Mini-ViT | 0.159 | 0.476 | 0.068 | 0.771 | 0.944 | 0.983 |
| LocalBins [19] | E-B5 | 0.156 | 0.470 | 0.067 | 0.777 | 0.949 | 0.985 |
| PixelFormer [20] | Swin-Large† | 0.144 | 0.441 | 0.062 | 0.802 | 0.962 | 0.990 |
| BinsFormer‡ [12] | Swin-Tiny | 0.162 | 0.478 | 0.069 | 0.760 | 0.945 | 0.985 |
| BinsFormer‡ [12] | Swin-Large† | 0.143 | 0.421 | 0.061 | 0.805 | 0.963 | 0.990 |
| **Ours** | Swin-Tiny | 0.157 | 0.476 | 0.069 | 0.768 | 0.950 | 0.987 |
| **Ours** | Swin-Large† | **0.135** | **0.405** | **0.059** | **0.822** | **0.971** | **0.993** |

Table 4: **Generalization to the SUN RGB-D dataset in a zero-shot setting with models trained on the NYU-Depth-v2 dataset.**

## 4.2 Implementation Details

Our framework is implemented in the PyTorch library [50] and trained on 4 NVIDIA A5000 24GB GPUs. The training process runs a total number of 20 epochs and takes around 24 hours. We utilize the Adam optimizer [51] and a batch size of 8. The learning rate is gradually reduced from 2e-5 to 2e-6 via the polynomial decay strategy.

## 4.3 Comparison to previous state-of-the-art competitors

**KITTI.** We report results on the Eigen split and official benchmark split, as summarized in Tables 1 and 2, respectively. For the Eigen split, the proposed method exceeds the leading approaches by a large margin, *e.g.*, compared to PixelFormer, which is also based on the classification-regression. In terms of the official benchmark split, the results are generated by the online server and the proposed method outperforms previous approaches again. It is worth noting that the main ranking metric SILog is reduced considerably.

In Fig. 4, we present qualitative depth comparison on the KITTI dataset. As we can see, the proposed method is capable of correctly predicting the depth of foreground objects, such as poles, even when the background is very complex, *e.g.*, lush foliage and bushes. In such complex scenes, PixelFormer and NeWCRFs are prone to foreground and background depth aliasing. Moreover, we notice that the compared methods, especially PixelFormer, tend to have mosaic-like artifacts at the top of the depth maps, but the proposed method does not.

**NYU-Depth-v2.** To further present the superiority of the proposed method in the indoor scenario, we evaluate it on the NYU-Depth-v2 dataset. The results are reported in Table 3, indicating that the proposed method surpasses prior competing approaches and achieves consistent improvements on most metrics. In particular, it improves BinsFormer by 7.4% and 4.8% on the Abs Rel and RMSE, respectively, which emphasizes the efficacy of our IEBins. In Fig. 5, we demonstrate qualitative depth

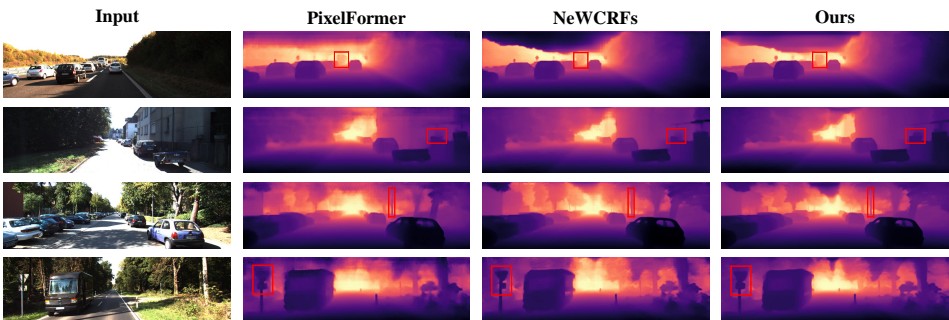

Figure 4: **Qualitative depth comparison on the KITTI dataset**. The red boxes show the regions to focus on.

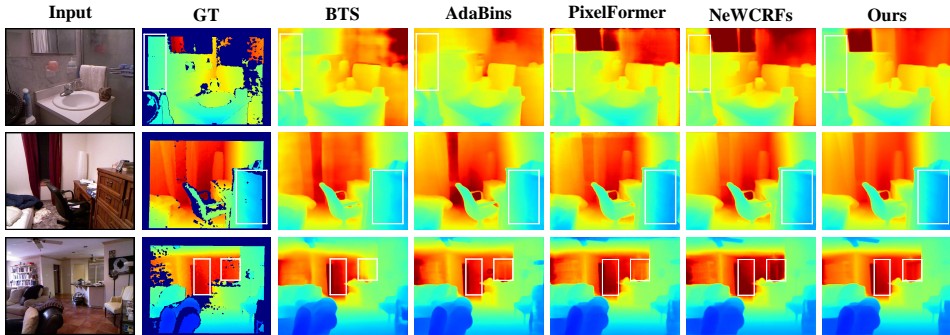

Figure 5: **Qualitative depth comparison on the NYU-Depth-v2 dataset**. The white boxes indicate the regions to focus on.

comparison. As can be seen, the proposed method preserves fine-grained details, such as boundaries and generates more continuous depth values in planar regions.

### 4.4 Zero-shot Generalization

Similar to previous approaches [6, 12], we conduct a cross-dataset evaluation in a zero-shot setting where the models are trained on the NYU-Depth-v2 dataset but evaluated on the SUN RGB-D dataset. As shown in Table 4, the proposed method achieves superior results than the compared approaches. Besides, we notice that the proposed method with Swin-Tiny backbone performs slightly worse than AdaBins on the NYU-Depth-v2 dataset, but on the SUN RGB-D it even surpasses AdaBins on some metrics like Abs Rel, which is an indicator of its excellent generalization ability.

### 4.5 Ablation Study

To better indicate the influence of each individual component, we conduct several ablation studies, divided into **IEBins**, **bin types** and **effect of bin numbers**.

**IEBins.** The importance of IEBins is first evaluated by comparing it with standard regression and IBins. We construct a baseline by removing the iterative optimizer from the whole framework. The standard regression is achieved by using the baseline to directly predict the depth map. The IBins represents that we replace the elastic target bin with the original target bin at each stage. Table 5 shows that the standard regression performs worse than IBins and IEBins. Benefiting from the elastic target bin, the more robust IEBins takes the IBins a step further.

**Bin types.** We then compare the IEBins against other choices including uniform bins [5], space increasing bins [5], adaptive bins [6] and local bins [19]. To make a fair comparison, we remain all other configurations the same except for bin types. As listed in Table 5, the baseline equipped with IEBins improves the performance markedly and surpasses other variants.

| Method | Abs Rel $\downarrow$ | RMSE $\downarrow$ | $\log_{10} \downarrow$ | $\delta < 1.25 \uparrow$ | $\delta < 1.25^2 \uparrow$ | $\delta < 1.25^3 \uparrow$ |
|---|---|---|---|---|---|---|
| Baseline + Regression | 0.095 | 0.337 | 0.041 | 0.921 | 0.991 | 0.998 |
| Baseline + UBins [5] | 0.091 | 0.328 | 0.040 | 0.925 | 0.991 | **0.998** |
| Baseline + SIBins [5] | 0.090 | 0.326 | 0.039 | 0.928 | **0.992** | 0.998 |
| Baseline + AdaBins [6] | 0.089 | 0.320 | **0.038** | 0.931 | 0.991 | **0.998** |
| Baseline + LocalBins [19] | 0.090 | 0.319 | **0.038** | 0.932 | **0.992** | 0.998 |
| Baseline + IBins | 0.090 | 0.317 | **0.038** | 0.932 | 0.991 | **0.998** |
| Baseline + IEBins | **0.087** | **0.314** | **0.038** | **0.936** | **0.992** | **0.998** |

Table 5: **Comparison of different bin types on the NYU-Depth-v2 dataset**. The baseline is built by removing the iterative optimizer from our whole framework. UBins: uniform bins; SIBins: space increasing bins; AdaBins: adaptive bins; LocalBins: local bins; IBins: iterative bins, which stands for replacing the elastic target bin with the original target bin at each stage. The number of bins for UBins, SIBins, AdaBins and LocalBins is set to 256 following [5, 6, 19].

| Number of Bins | Abs Rel $\downarrow$ | RMSE $\downarrow$ | $\log_{10} \downarrow$ | $\delta < 1.25 \uparrow$ | $\delta < 1.25^2 \uparrow$ | $\delta < 1.25^3 \uparrow$ |
|---|---|---|---|---|---|---|
| 4 | 0.105 | 0.335 | 0.043 | 0.900 | 0.982 | 0.999 |
| 8 | 0.089 | 0.317 | **0.038** | 0.934 | **0.992** | **0.998** |
| 16 | **0.087** | **0.314** | **0.038** | **0.936** | 0.992 | 0.998 |
| 32 | 0.088 | 0.317 | **0.038** | 0.932 | **0.992** | **0.998** |

Table 6: **Effect of bin numbers on the NYU-Depth-v2 dataset**.

| Method | Backbone | Abs Rel $\downarrow$ | Parameters (M) $\downarrow$ | Inference Time (s) $\downarrow$ |
|---|---|---|---|---|
| NeWCRFs [8] | Swin-Large† | 0.095 | 270 | 0.052 |
| BinsFormer‡ [12] | Swin-Large† | 0.094 | 255 | 0.216 |
| Ours | Swin-Large† | 0.087 | 273 | 0.085 |

Table 7: **Comparison of model parameters and inference time on the NYU-Depth-v2 dataset**.

**Effect of bin numbers.** To examine how the number of bins affects the performance, we train our model using different values of the bin number. The results are reported in Table 6. The error drops sharply as the number of bins increases, and then this drop disappears when it is greater than 16 bins. Therefore, we use 16 bins in our final model.

### 4.6 Model Parameters and Inference Time

We conduct a comparison between the proposed method, NeWCRFs and BinsFormer based on their inference time and the number of model parameters in Table 7, with the Swin-Large backbone. We measure the inference time on the NYU-Depth-v2 test set with a batch size of 1. It can be seen that the number of parameters of the proposed method is almost equal to that of NeWCRFs and slightly more than that of BinsFormer. Nevertheless, the proposed method is nearly $60\%$ faster than BinsFormer in inference time, also as the classification-regression-based method. Meanwhile, the proposed method achieves much better performance than these two counterparts. Hence, the proposed method provides a better balance between performance, number of parameters and inference time.

## 5 Limitations

We acknowledge the following limitations: First, we use the classification-regression to predict depth. Compared with the classification, the depth acquired by classification-regression is smoother and more continuous, but at the same time it may blur the boundaries due to the weighted average of all depth candidates. Second, we use the pixel-wise loss to supervise depth without imposing a direct supervision signal on the probabilistic distribution. This does not guarantee that the depth candidate corresponding to the peak point of the probability distribution is the real optimal depth candidate, thereby affecting the depth estimate.

# 6 Conclusion

In this paper, we introduce a novel concept of iterative elastic bins for the classification-regression-based monocular depth estimation. The proposed iterative elastic bins uses multiple small number of bins to progressively search for high-quality depth and can be plugged into other frameworks as a strong baseline. In addition, we present a dedicated framework composed of a feature extractor and an iterative optimizer. The performance is evaluated on two popular datasets from both outdoor and indoor scenarios and the proposed method exceeds previous state-of-the-art competitors. Furthermore, its generalization ability is verified in a zero-shot setting on the SUN RGB-D dataset.

**Acknowledgments**

This work was supported in part by the National Natural Science Foundation of China under grant U1909215 and 51975029, in part by the A*STAR Singapore through Robotics Horizontal Technology Coordinating Office (HTCO) under Project C221518005, in part by the Key Research and Development Program of Zhejiang Province under Grant 2021C03050, in part by the Scientific Research Project of Agriculture and Social Development of Hangzhou under Grant No. 20212013B11, and in part by the National Natural Science Foundation of China under grant 61620106012 and 61573048.

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
