# Supplementary Material
# IEBins: Iterative Elastic Bins for
# Monocular Depth Estimation

**Shuwei Shao[1], Zhongcai Pei[1], Xingming Wu[1], Zhong Liu[1], Weihai Chen[2,*] and Zhengguo Li[3]**

[1]School of Automation Science and Electrical Engineering, Beihang University, China
[2]School of Electrical Engineering and Automation, Anhui University, China
[3]SRO department, Institute for Infocomm Research, A*STAR, Singapore
*Corresponding author, email:whchen@buaa.edu.cn

## 1    More Ablation Results

To better understand the influence of each individual component on the KITTI dataset, we provide results in Tables 1 and 2. As we can see from Table 1, the proposed IEBins exceeds other counterparts again. Table 2 shows a similar performance trend as in NYU-Depth-v2 dataset with increasing number of bins. Besides, we provide results of different training stage numbers on the NYU-Depth-v2 dataset in Table 3. As the stage increases, the performance gradually improves until saturated, and when the number of stages exceeds 6, the performance changes very little.

| Method | Abs Rel ↓ | Sq Rel ↓ | RMSE ↓ | RMSE log ↓ | $\delta < 1.25$ ↑ | $\delta < 1.25^2$ ↑ | $\delta < 1.25^3$ ↑ |
|---|---|---|---|---|---|---|---|
| Baseline + Regression | 0.053 | 0.153 | 2.129 | 0.080 | 0.974 | 0.997 | 0.999 |
| Baseline + UBins [1] | 0.052 | 0.151 | 2.095 | 0.078 | 0.975 | 0.997 | 0.999 |
| Baseline + SIBins [1] | 0.051 | 0.147 | 2.092 | 0.078 | 0.976 | 0.997 | 0.999 |
| Baseline + AdaBins [2] | 0.051 | 0.146 | 2.048 | 0.077 | 0.976 | **0.998** | **0.999** |
| Baseline + IBins | **0.050** | 0.143 | 2.050 | 0.076 | 0.977 | **0.998** | **0.999** |
| Baseline + IEBins | **0.050** | **0.142** | **2.011** | **0.075** | **0.978** | **0.998** | **0.999** |

Table 1: **Comparison of different bin types on the KITTI dataset**.

| Number of Bins | Abs Rel ↓ | Sq Rel ↓ | RMSE ↓ | RMSE log ↓ | $\delta < 1.25$ ↑ | $\delta < 1.25^2$ ↑ | $\delta < 1.25^3$ ↑ |
|---|---|---|---|---|---|---|---|
| 4 | 0.288 | 1.199 | 4.146 | 0.277 | 0.574 | 0.910 | 0.979 |
| 8 | 0.054 | 0.147 | **2.009** | 0.080 | 0.973 | 0.996 | **0.999** |
| 16 | **0.050** | **0.142** | 2.011 | **0.075** | **0.978** | **0.998** | **0.999** |
| 32 | **0.050** | **0.142** | 2.026 | 0.076 | 0.977 | **0.998** | **0.999** |

Table 2: **Effect of bin numbers on the KITTI dataset**.

## 2    More Qualitative Results

We present qualitative depth comparison on the official split of KITTI dataset in Fig. 1. It can be seen that the proposed method is more capable of delineating difficult object boundaries, *e.g.*, human. To further evaluate the depth quality from the 3D shape, we convert the depth maps into point clouds and present qualitative point cloud comparison on the KITTI and NYU-Depth-v2 datasets in Figs. 2 and 3, respectively. As can be seen, the proposed method shows less distortion than the compared approaches and recovers the structures of the 3D world reasonably.

37th Conference on Neural Information Processing Systems (NeurIPS 2023).

| Number of Stages | Abs Rel ↓ | RMSE ↓ | $\log_{10}$ ↓ | $\delta < 1.25$ ↑ | $\delta < 1.25^2$ ↑ | $\delta < 1.25^3$ ↑ |
|:---:|:---:|:---:|:---:|:---:|:---:|:---:|
| 1 | 0.093 | 0.333 | 0.041 | 0.921 | 0.991 | **0.998** |
| 2 | 0.090 | 0.325 | 0.040 | 0.927 | 0.991 | **0.998** |
| 3 | 0.089 | 0.320 | 0.039 | 0.931 | 0.991 | **0.998** |
| 4 | 0.088 | 0.317 | **0.038** | 0.933 | **0.992** | **0.998** |
| 5 | **0.087** | 0.315 | **0.038** | 0.935 | **0.992** | **0.998** |
| 6 | **0.087** | 0.314 | **0.038** | **0.936** | **0.992** | **0.998** |
| 7 | **0.087** | **0.313** | **0.038** | 0.935 | **0.992** | **0.998** |

Table 3: **Effect of training stage numbers on the NYU-Depth-v2 dataset**.

# 3 Application to SLAM

To exhibit the benefits of improvements in downstream tasks such as SLAM, we integrate IEBins and NeWCRFs [3] into ORB-SLAM2 [4] in the RGB-D setting and evaluate the visual odometry performance on the KITTI odometry dataset. We report results on keyframes (selected by the ORB-SLAM2) and on all frames of sequences 01-10. The ATE (m) metric is used. As shown in Table 4, the proposed method either significantly exceeds the NeWCRFs or achieves on par performance with the latter.

| Sequence | IEBins (key) | NeWCRFs (key) | IEBins (all) | NeWCRFs (all) |
|:---:|:---:|:---:|:---:|:---:|
| 01 | 117.06 | 536.53 | 125.09 | 583.20 |
| 02 | 12.22 | 13.32 | 13.59 | 13.97 |
| 03 | 6.72 | 8.31 | 7.15 | 9.04 |
| 04 | 16.70 | 31.56 | 16.61 | 30.59 |
| 05 | 8.10 | 8.05 | 7.56 | 7.86 |
| 06 | 1.32 | 0.96 | 1.35 | 0.95 |
| 07 | 2.48 | 3.09 | 2.55 | 3.24 |
| 08 | 10.89 | 9.82 | 11.06 | 9.90 |
| 09 | 5.44 | 7.61 | 5.68 | 7.67 |
| 10 | 7.21 | 11.73 | 8.24 | 12.66 |

Table 4: **Visual odometry results on the KITTI odometry dataset**. "key" and "all" stand for keyframes and all frames, respectively.

# 4 Quantitative Evidence Towards the Working of IEBins

We randomly choose a sample from the NYU-Depth-v2 test set, and show the median elastic target bin width across the image, corresponding uncertainty values and elasticity factors (new adjusted width divided by the bin-width at that stage with no elasticity) for each stage. The results are listed in Table 5. It can be seen that as the stage increases, the elastic target bin widths and uncertainty values continue to decrease. The elasticity factors are between 3.9 and 4.8.

| | Stage 1 | Stage 2 | Stage 3 | Stage 4 | Stage 5 | Stage 6 |
|:---:|:---:|:---:|:---:|:---:|:---:|:---:|
| Width (median) | 9.9 | 2.561 | 0.756 | 0.228 | 0.073 | 0.024 |
| Uncertainty (std) | - | 0.971 | 0.281 | 0.087 | 0.029 | 0.010 |
| Elasticity factor | - | 4.139 | 3.917 | 4.222 | 4.562 | 4.800 |

Table 5: **Median elastic target bin width across the image, corresponding uncertainty values and elasticity factors for different stages**.

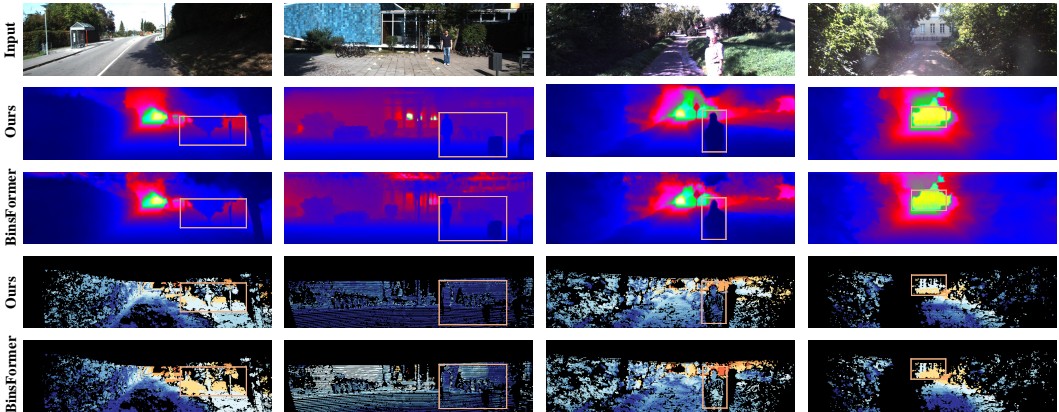

Figure 1: **Qualitative depth comparison on the KITTI online benchmark**. The results are generated by the online server. The second and third rows stand for depth predictions and the fourth and fifth rows stand for corresponding error maps, where the large errors are in orange or red. The orange boxes indicate the regions to emphasize.

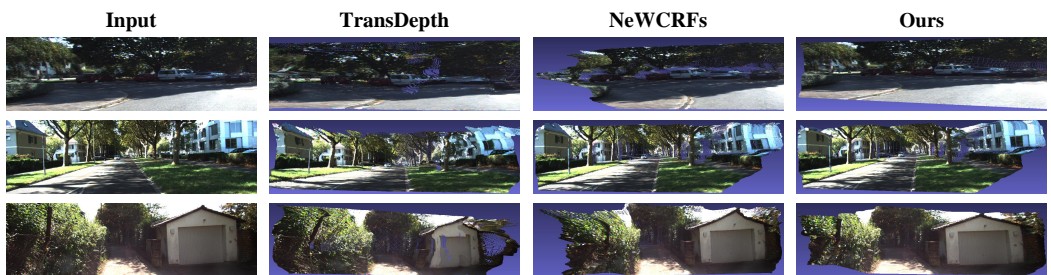

Figure 2: **Qualitative point cloud comparison on the KITTI dataset**.

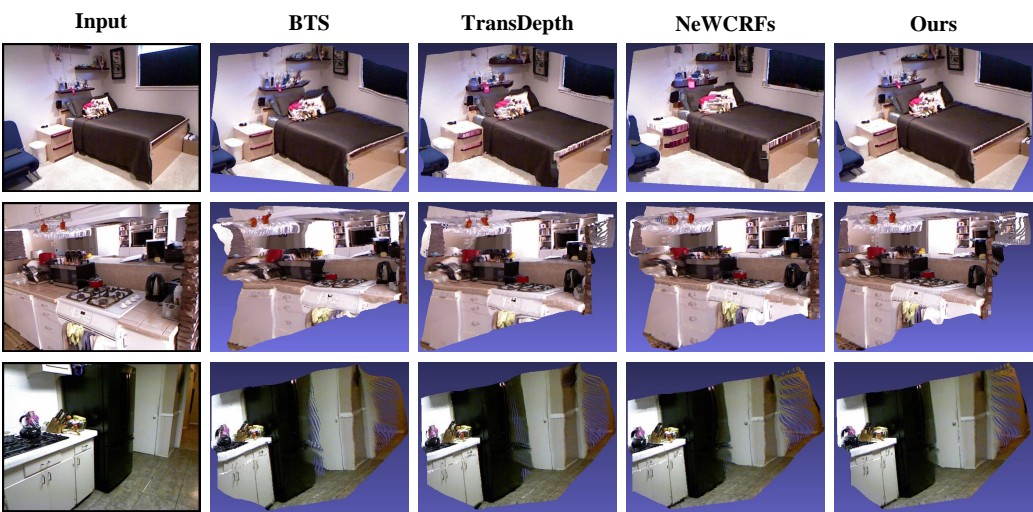

Figure 3: **Qualitative point cloud comparison on the NYU-Depth-v2 dataset**.