# OpenReview forum: "IEBins: Iterative Elastic Bins for Monocular Depth Estimation"
_NeurIPS.cc/2023/Conference — NeurIPS 2023 poster_

### Official Review · Reviewer_TtXH · 2023-07-02

**Soundness:** 3 good
**Presentation:** 3 good
**Contribution:** 2 fair
**Rating:** 5
**Confidence:** 4

**Summary:**

The paper introduces an Iterative Elastic Bins (IEBins) approach for monocular depth estimation. Many conventional monocular depth estimation approach uses a soft-argmax representation (Eq. (4)) that sums the product between depth probability and pre-defined depth bins. However, the large number of pre-defined bins hinders model convergence (@L.43). To alleviate the problem, the paper proposes to use adaptive ranges of depth bins for each pixel based on previous intermediate depth estimates. Along with the number of iteration steps, the depth ranges get smaller for fine-detail estimation (yet, taking uncertainty into account). The paper shows empirically good accuracy over previous methods.

**Strengths:**

+ Good results

  Table 1, 2, 3, and 4 show that the method achieves better accuracy than published methods in the public benchmark datasets (KITTI Eigen, KITTI, NYU-Depth-v2, and SUN RGB-D).

+ Good ablation study

   Table 5 and 6 demonstrates how each design choice affects/improves the accuracy. Table 5 shows the strength of the proposed IEBin idea over previous approaches (UBins, SIBins, AdaBins, and IBins). Table 6 also justifies the design choice of the number of bins.

**Weaknesses:**

- Question on the number of stages (Fig. 3)

  The method uses 6 stages (iteration steps) in total. I wonder what happens if it uses more iteration steps during the inference (i.e., 6 iterations during training, but 8 or 10 iterations during test time). Will the method keep improving the accuracy with more iteration steps? Or, even during training, what if it uses more iteration steps other than 6? How the number 6 is set?

- Possibly an unfair comparison? Table 5, Comparison with different bin types.

  I wonder if other methods (UBins, SIBins, AdaBins, and IBins) also use the iterative optimizer. For a fair comparison, I think it's also good to prepare a baseline with the iterative optimizer and only change the depth bin types. Then it can differentiate the performance gain of bin types from the iterative optimizer.


Despite the good results, I would like the paper Borderline Reject for now, mainly due to the possible unfair comparison in Table 5. The source of gain is not so clear if it's from the iterative optimizer or the depth bin types.

----

I share the same concern with Reviewer 4WPj that the main contributions don't seem to be so strong. The iterative refinement ideas have been proposed in other literature (e.g., optical flow, SfM), and the paper demonstrates empirical accuracy gain. That said, I am raising the score to Borderline Accept as the authors' response resolves my main concerns. I hope all the comments and concerns during the discussion phase are reflected in the updated version.

**Questions:**

- In Fig. 2 the three depth maps on the right look the same. Probably it may need to be changed to the actual experiment results.
- In the last example (chair) in Fig. 5, the object boundary near the chair is still blurry. (Just out of curiosity) I wonder if the proposed representation can resolve this blurry object boundary problem because it limits the depth range, so I would expect it outputs much clearer depth around object boundaries.
- In Table 7, what is the source of the faster inference time than BinsFormer?

**Limitations:**

The paper didn't include its limitations or societal impact.

---

> ### Author Rebuttal · Authors · 2023-08-07
>
> ### __We thank our reviewer for the constructive feedback and comments.__
> ### _W1: Question on the number of stages (Fig. 3)_
>
> A1: During both training and inference phases, we find that when the number of stages exceeds 6, the performance changes very little. As we know, more stages require longer training and inference time, greater memory consumption. Hence, we set the number of stages to 6.
>
> ### _W2: Possibly an unfair comparison? Table 5, Comparison with different bin types._
>
> A2: We apologize for not stating the settings clearly. Because UBins, SIBins and AdaBins are non-iterative methods, we do not add iterative optimizer for them. On the other hand, Baseline + AdaBins (276M) has more parameters than Baseline + IEBins (273M) because AdaBins requires an additional Transformer architecture to generate adaptive bins. IBins is a variant of IEBins and is acquired by replacing the elastic target bin with original target bin at each stage. We have used iterative optimizer for IBins when reporting its results in Table 5. To further elaborate results, we take well-based AdaBins as an example and feed its adaptive depth candidates into iterative optimizer. The corresponding results of Abs Rel, RMSE, log10, $\delta <1.25$, $\delta <1.25^2$, $\delta <1.25^3$ are 0.089, 0.321, 0.038, 0.932, 0.991, and 0.998 on the NYU-Depth-v2 dataset, which are worse than those of Baseline + IEBins (0.087, 0.314, 0.038, 0.936, 0.992, and 0.998).
>
> ### _Q1: Probably it may need to be changed to the actual experiment results for three depth maps in Fig.2._
>
> A3: We will modify our paper according to this nice advice.
>
> ### _Q2: In the last example (chair) in Fig. 5, the object boundary near the chair is still blurry. (Just out of curiosity) I wonder if the proposed representation can resolve this blurry object boundary problem because it limits the depth range, so I would expect it outputs much clearer depth around._
>
> A4: Yes, this is possible because the proposed representation of iterative division of bins is sensitive to the object boundaries due to large depth variations in these regions.
>
> ### _Q3: In Table 7, what is the source of the faster inference time than BinsFormer?_
>
> A5: BinsFormer contains two decoders, a pixel decoder and a transformer decoder, and frequent interactions occur between these two decoders, which may increase the inference time significantly. Although our method is iterative, the developed iterative optimizer is lightweight and operates at 1/4 resolution.

---

> > ### Comment · Reviewer_TtXH · 2023-08-13
> > **Discussion**
> >
> > Thanks for sharing your responses!
> >
> > However, there are some unclear thoughts after reading the response.
> >
> > > W1: Question on the number of stages (Fig. 3)
> >
> > What does it mean by that ``` When the number of stages exceeds 6, the performance changes very little.```? Does it mean that the accuracy has plateaued? It would have been much great if the paper provided an analysis on trying out different numbers of iteration steps during both training and testing (even trying out different training/testing numbers, e.g., training 6 iterations, testing 8 iterations).
> >
> >
> > > Q2: In the last example (chair) in Fig. 5, the object boundary near the chair is still blurry.
> >
> > I am sorry but I didn't understand the answer clearly. Could you elaborate more on why ```the proposed method is sensitive to the object boundaries due to large depth variations in those regions```?

---

> > > ### Author Response · Authors · 2023-08-14
> > > **Discussion**
> > >
> > > ### __Thank you very much for your feedback.__
> > >
> > > ### _W1: What does it mean by that when the number of stages exceeds 6, the performance changes very little.? Does it mean that the accuracy has plateaued? It would have been much great if the paper provided an analysis on trying out different numbers of iteration steps during both training and testing (even trying out different training/testing numbers, e.g., training 6 iterations, testing 8 iterations)._
> > >
> > > A1:  As suggested, we have provided the results of different stages below (the number of stages for training and inference remains the same).
> > >
> > > |Stage &nbsp; Abs Rel &nbsp; RMSE &nbsp; log10 &nbsp; $\delta <1.25$ &nbsp; $\delta <1.25^2$ &nbsp; $\delta <1.25^3$||
> > > |:---|:---|
> > > |Stage1 &nbsp; 0.093 &ensp;&ensp; 0.333 &ensp; 0.041 &ensp;&ensp;&nbsp; 0.921 &ensp;&ensp;&ensp; 0.991 &ensp;&ensp;&ensp;&ensp;&ensp; 0.998|
> > > |Stage2 &nbsp; 0.090 &ensp;&ensp; 0.325 &ensp; 0.040 &ensp;&ensp;&nbsp; 0.927 &ensp;&ensp;&ensp; 0.991 &ensp;&ensp;&ensp;&ensp;&ensp; 0.998|
> > > |Stage3 &nbsp; 0.089 &ensp;&ensp; 0.320 &ensp; 0.039 &ensp;&ensp;&nbsp; 0.931 &ensp;&ensp;&ensp; 0.991 &ensp;&ensp;&ensp;&ensp;&ensp; 0.998|
> > > |Stage4 &nbsp; 0.088 &ensp;&ensp; 0.317 &ensp; 0.038 &ensp;&ensp;&nbsp; 0.933 &ensp;&ensp;&ensp; 0.992 &ensp;&ensp;&ensp;&ensp;&ensp; 0.998|
> > > |Stage5 &nbsp; 0.087 &ensp;&ensp; 0.315 &ensp; 0.038 &ensp;&ensp;&nbsp; 0.935 &ensp;&ensp;&ensp; 0.992 &ensp;&ensp;&ensp;&ensp;&ensp; 0.998|
> > > |Stage6 &nbsp; 0.087 &ensp;&ensp; 0.314 &ensp; 0.038 &ensp;&ensp;&nbsp; 0.936 &ensp;&ensp;&ensp; 0.992 &ensp;&ensp;&ensp;&ensp;&ensp; 0.998|
> > > |Stage7 &nbsp; 0.087 &ensp;&ensp; 0.313 &ensp; 0.038 &ensp;&ensp;&nbsp; 0.935 &ensp;&ensp;&ensp; 0.992 &ensp;&ensp;&ensp;&ensp;&ensp; 0.998|
> > >
> > > As the stage increases, the performance gradually improves until saturated, and when the number of iterations exceeds 6, the performance changes very little.
> > >
> > > Then we try to train using 6 stages and use more stages in inference phase. The results are as follows:
> > >
> > > |Stage &nbsp; Abs Rel &nbsp; RMSE &nbsp; log10 &nbsp; $\delta <1.25$ &nbsp; $\delta <1.25^2$ &nbsp; $\delta <1.25^3$||
> > > |:---|:---|
> > > |Stage6 &nbsp; 0.087 &ensp;&ensp; 0.314 &ensp; 0.038 &ensp;&ensp;&nbsp; 0.936 &ensp;&ensp;&ensp; 0.992 &ensp;&ensp;&ensp;&ensp;&ensp; 0.998|
> > > |Stage7 &nbsp; 0.087 &ensp;&ensp; 0.314 &ensp; 0.038 &ensp;&ensp;&nbsp; 0.936 &ensp;&ensp;&ensp; 0.992 &ensp;&ensp;&ensp;&ensp;&ensp; 0.998|
> > > |Stage8 &nbsp; 0.087 &ensp;&ensp; 0.314 &ensp; 0.038 &ensp;&ensp;&nbsp; 0.936 &ensp;&ensp;&ensp; 0.992 &ensp;&ensp;&ensp;&ensp;&ensp; 0.998|
> > >
> > > As we can see, the performance becomes plateaued when using 7 stages and 8 stages at inference phase in this case.
> > >
> > > ### _Q2: Could you elaborate more on why the proposed method is sensitive to the object boundaries due to large depth variations in those regions?_
> > >
> > > A2: Generally, the depth variations are large at object boundaries. In these regions, due to the large depth variations, the proposed method can easily classify respective region into different depth ranges at initialization stage and further iteratively refine these depth ranges in subsequent stages. Hence, it is possible for the proposed representation to solve this blurry object boundary problem.

---

> > > > ### Comment · Reviewer_TtXH · 2023-08-14
> > > >
> > > > Thanks for providing the detailed analysis! It resolved my concern.
> > > >
> > > > > Q2: Could you elaborate more on why the proposed method is sensitive to the object boundaries due to large depth variations in those regions?
> > > >
> > > > I see. My original question is that why the proposed method still give blurry object boundaries on the chair example (Fig. 5, the bottom row) although it's expected to solve this problem via the iterative refinement? In Fig. 5, the proposed method gives sharper estimation than at least other methods, but (in my humble, subjective opinion) the depth boundaries still look blurry. I am not trying to pick on this single example, but trying to understand the upper bound performance of this method -- what the method can solve and cannot solve. Any discussion on this would be appreciated!

---

> > > > > ### Author Response · Authors · 2023-08-15
> > > > >
> > > > > ### __Thank you very much for your prompt feedback.__
> > > > >
> > > > > ### _Q2: My original question is that why the proposed method still give blurry object boundaries on the chair example (Fig. 5, the bottom row) although it's expected to solve this problem via the iterative refinement? In Fig. 5, the proposed method gives sharper estimation than at least other methods, but (in my humble, subjective opinion) the depth boundaries still look blurry. I am not trying to pick on this single example, but trying to understand the upper bound performance of this method -- what the method can solve and cannot solve. Any discussion on this would be appreciated!_
> > > > >
> > > > > A1: Nice comments. We think there are two possible reasons:
> > > > >
> > > > > First, we use the classification-regression to estimate depth. Compared with the classification, the depth acquired by classification-regression is smoother and more continuous, but at the same time it may blur the boundaries due to the weighted average of all depth candidates.
> > > > >
> > > > > Second, we use a pixel-wise loss to supervise the depth without imposing a direct supervision signal on the probabilistic distribution. This does not guarantee that the depth candidate corresponding to the peak point of the probability distribution is the true optimal depth candidate, thereby affecting the depth estimate.

---

> > > > > > ### Comment · Reviewer_TtXH · 2023-08-15
> > > > > >
> > > > > > Thanks for the comments! I hope this discussion will be included in the main paper or limitation section.

---

> > > > > > > ### Author Response · Authors · 2023-08-16
> > > > > > >
> > > > > > > ### __Thank you for the feedback.__
> > > > > > >
> > > > > > > We will include the above discussion in the main paper or limitation section as suggested.

---

### Official Review · Reviewer_jxnL · 2023-07-04

**Soundness:** 4 excellent
**Presentation:** 4 excellent
**Contribution:** 4 excellent
**Rating:** 8
**Confidence:** 2

**Summary:**

The paper introduces a method for monocular depth. The method uses a recurrent network based on RAFT to predict a probability distribution over a set of bins, which enables the depth at each iteration to be computed as the expectation over all bins and the margins of each bin to be adjusted using the computed variance. In the subsequent iterations, a finer-grained search is performed in the adjusted bins. This new strategy is introduced as "IEBins". The authors motivate this strategy as being robust to uncertain initial predictions.

**Strengths:**

The method outperforms prior work across multiple standard evaluation settings.

The method is interpretable, producing confidence for different regions and steadily improving depth predictions.

To the best of my knowledge, the iterative design is substantially different than the usual approach for monocular depth.

The introduced method is very fast.

The IEBins are compared against other binning strategies in ablation experiments.

**Weaknesses:**

No obvious weaknesses.

**Questions:**

None

**Limitations:**

I did not see discussion of limitations

---

> ### Author Rebuttal · Authors · 2023-08-07
>
> ### __We thank our reviewer very much for the highly positive feedback.__

---

### Official Review · Reviewer_LZ7t · 2023-07-06

**Soundness:** 2 fair
**Presentation:** 2 fair
**Contribution:** 2 fair
**Rating:** 4
**Confidence:** 4

**Summary:**

This paper introduces a classification-regression-based monocular depth estimation pipeline.
Previous classification or classification-regression-based MDE approaches often suffer from huge complexity and loss of generalizability issue, due to the nature of requiring more depth hypothesis for better performance.
In this paper, it propose iterative elastic bins, ie, IEBins technique, which iteratively adjusts bin ranges as the prediction converges, so that with less number of bins and fast inference time, the method can still perform on the state-of-the-art level.
It additionally introduces dedicated transformer-based feature extractor and GRU-based iterative optimizer.

**Strengths:**

This paper is well-written and easy to follow.
The proposed IEBins is an intuitive and interesting idea, and the performance also backs up its effectiveness.

**Weaknesses:**

More in-depth analysis on 'iterative' perspective of IEBins will be appreciated.

1) For example, what happens when iteration is less or more than 6? How much does the number of iterations affect the final performance?

2) The authors claim that the proposed GRU-based iterative optimizer is also one of their contributions, saying that it is helpful to capture temporal information during IEBins-based depth estimation. Yet, there is no ablation study regarding this. Comparison between adding/excluding GRU unit, alongside its effectiveness regarding the number of iteration will help strengthening the claimed contribution.

**Questions:**

1. I have some questions regarding updating rules of the proposed IEBins.

1-1. In Eq. 7, it denotes that elastic target bin edges are modified using uncertainty from previous stage's probability distribution. Does it mean that for each pixel, bin edges are set differently? If this is the case, the network should remember separate bin ranges for every pixel, which seems like to require huge memory. Correct me if I'm wrong.

1-2. In L133, the paper says that d_min and d_max values are updated with new bin edges. In the next iteration, are 16 bins additionally set within this new min-max range? Then, does it mean that in 2nd iteration, for example, new bin range are approximately 1/256 from original min-max range?

2. In Fig 5, the range of the proposed method seems significantly off compared to GT. How is the visualization done?

3. In Tab 6, why does the performance drop when using 32 bins instead of 16? Is this aligned with overfitting issue stated in the introduction section? If it's true, than doesn't it mean that the proposed method already suffers from 32 bins while AdaBins, for example, can operate until 256 bins? How can this phenomenon be analyzed?

**Limitations:**

The authors did not address any limitations. Yet, other than weakness and questions I commented above, I don't see any severe limitation to this paper.

---

> ### Author Rebuttal · Authors · 2023-08-07
>
> ### __We thank our reviewer for the constructive feedback and comments.__
>
> ### _W1: What happens when iteration is less or more than 6? How much does the number of iterations affect the final performance?_
>
> A1: The results of IEBins at different stages are shown below:
> |Stage  &nbsp;  Abs Rel  &nbsp;   RMSE  &nbsp;  log10  &nbsp;   $\delta  < 1.25$  &nbsp;   $\delta  < {1.25^2}$ &nbsp; $\delta <1.25^3$||
> |:---|:---|
> |Stage1 &nbsp; 0.093 &emsp; 0.333 &ensp; 0.041 &emsp; 0.921 &emsp;&ensp;&ensp; 0.991 &emsp;&ensp;&ensp; 0.998|
> |Stage2 &nbsp; 0.090 &emsp; 0.325 &ensp; 0.040 &emsp; 0.927 &emsp;&ensp;&ensp; 0.991 &emsp;&ensp;&ensp; 0.998|
> |Stage3 &nbsp; 0.089 &emsp; 0.320 &ensp; 0.039 &emsp; 0.931 &emsp;&ensp;&ensp; 0.991 &emsp;&ensp;&ensp; 0.998|
> |Stage4 &nbsp; 0.088 &emsp; 0.317 &ensp; 0.038 &emsp; 0.933 &emsp;&ensp;&ensp; 0.992 &emsp;&ensp;&ensp; 0.998|
> |Stage5 &nbsp; 0.087 &emsp; 0.315 &ensp; 0.038 &emsp; 0.935 &emsp;&ensp;&ensp; 0.992 &emsp;&ensp;&ensp; 0.998|
> |Stage6 &nbsp; 0.087 &emsp; 0.314 &ensp; 0.038 &emsp; 0.936 &emsp;&ensp;&ensp; 0.992 &emsp;&ensp;&ensp; 0.998|
> |Stage7 &nbsp; 0.087 &emsp; 0.313 &ensp; 0.038 &emsp; 0.935 &emsp;&ensp;&ensp; 0.992 &emsp;&ensp;&ensp; 0.998|
>
> As the stage increases, the performance gradually improves until saturated, and when the number of iterations exceeds 6, the performance changes very little.
>
> ### _W2: Ablation study on the GRU-based iterative optimizer._
>
> A2:  Nice comment. We have presented the results at each stage after excluding GRU unit from the iterative optimizer below. For results of the whole framework, please refer to A1 to W1.
>
> |Stage  &nbsp;  Abs Rel  &nbsp;   RMSE  &nbsp;  log10  &nbsp;   $\delta  < 1.25$  &nbsp;   $\delta  < {1.25^2}$ &nbsp; $\delta <1.25^3$||
> |:---|:---|
> |Stage1 &nbsp; 0.093 &emsp; 0.334 &ensp; 0.041 &emsp; 0.920 &emsp;&ensp;&ensp; 0.991 &emsp;&ensp;&ensp; 0.998|
> |Stage2 &nbsp; 0.091 &emsp; 0.327 &ensp; 0.040 &emsp; 0.925 &emsp;&ensp;&ensp; 0.991 &emsp;&ensp;&ensp; 0.998|
> |Stage3 &nbsp; 0.090 &emsp; 0.323 &ensp; 0.039 &emsp; 0.928 &emsp;&ensp;&ensp; 0.991 &emsp;&ensp;&ensp; 0.998|
> |Stage4 &nbsp; 0.089 &emsp; 0.320 &ensp; 0.039 &emsp; 0.930 &emsp;&ensp;&ensp; 0.991 &emsp;&ensp;&ensp; 0.998|
> |Stage5 &nbsp; 0.088 &emsp; 0.318 &ensp; 0.039 &emsp; 0.931 &emsp;&ensp;&ensp; 0.992 &emsp;&ensp;&ensp; 0.998|
> |Stage6 &nbsp; 0.088 &emsp; 0.317 &ensp; 0.039 &emsp; 0.932 &emsp;&ensp;&ensp; 0.992 &emsp;&ensp;&ensp; 0.998|
>
> The results verify the efficacy of the GRU-based iterative optimizer.
>
> ### _Q1.1: Confusion over bin edges._
>
> A3: The bin edges vary between pixels besides the initialization stage. The iterative optimizer operates at 1/4 resolution in our design, making the memory consumption affordable.
>
> ### _Q1.2: Are 16 bins additionally set within this new min-max range? And in 2nd iteration, new bin range are approximately 1/256 from original min-max range?_
>
> A4: Yes, 16 bins are set within the target bin by using the target bin as a new min-max range at each stage. For the case of target bin, the new bin range are 1/256 from original min-max range in 2nd stage. While for the case of elastic target bin, this ratio varies from pixel to pixel and image to image due to the adaptive nature introduced by elastic target bin.
>
> ### _Q2: How is the visualization done?_
>
> A5: We scale each depth map individually here to acquire the colormap. The GT depth maps are not completely dense and have missing values in border regions, which may affect the final colormap.
>
> ### _Q3: In Tab 6, why does the performance drop when using 32 bins instead of 16? Is this aligned with overfitting issue stated in the introduction section?_
>
> A6: This phenomenon may not be induced by the overfitting issue. As the number of bins increases, it becomes more and more difficult to classify the true optimal candidate for the next stage from a larger number of depth candidates. When the target bin classified deviates so far from the true/GT depth that the true depth also cannot be included in the elastic target bin, large depth errors will be generated in subsequent stages, thereby affecting the final performance.

---

> > ### Comment · Reviewer_LZ7t · 2023-08-14
> >
> > I thank the authors for their rebuttal.
> >
> > W2. The performance boost obtained from the GRU-based optimizer is around 0.001-0.005, RMSE-wise, which is not significantly high. Also, it degrades results on earlier stages of iterations. Can authors provide more insight or arguments regarding this?
> >
> > Q2. Minor note. I think it would be more effective to set the min-max normalizing values same throught a single sample. From what is given now, it is hard to know if one only preserves fine-grained detail or it actually predicts precise relative depth map.

---

> > > ### Author Response · Authors · 2023-08-14
> > > **Discussion**
> > >
> > > ### __Thank you very much for your feedback.__
> > > ### _W2: The performance boost obtained from the GRU-based optimizer is around 0.001-0.005, RMSE-wise, which is not significantly high. Also, it degrades results on earlier stages of iterations. Can authors provide more insight or arguments regarding this?_
> > >
> > > ### _W2.1: The performance boost obtained from the GRU-based optimizer is around 0.001-0.005, RMSE-wise, which is not significantly high._
> > >
> > > A1: As we can see from the results, the advantage of the GRU-based iterative optimizer is not obvious when the number of stages is 1, which may be due to the fact that there is no historical hidden state available. As the number of stages increases, the advantage of the GRU-based iterative optimizer gradually increases and tends to be stable. The GRU unit we use in our iterative optimizer is very lightweight and only contains three separable 5x5 convolution kernels. As for the number of parameters introduced, the performance gains brought by the GRU unit is competitive.
> > >
> > > ### _W2.2: Also, it degrades results on earlier stages of iterations. Can authors provide more insight or arguments regarding this?_
> > >
> > > A2: When the stage is 7, the depth candidates are highly close. In this case, it is very difficult for the iterative optimizer to classify the true optimal depth candidate and it is easy to make mistakes, which may cause slight performance degradation.
> > >
> > > ### _Q2: Minor note. I think it would be more effective to set the min-max normalizing values same throught a single sample. From what is given now, it is hard to know if one only preserves fine-grained detail or it actually predicts precise relative depth map._
> > >
> > > A3: We will revise our paper according to this nice advice.

---

> > > > ### Comment · Reviewer_LZ7t · 2023-08-21
> > > >
> > > > I thank the authors for their response.
> > > > While I think the IEBins design itself is noteworthy, I'm not totally convinced by additional contributions (such as GRU-based optimizer) and analysis.
> > > > Thus, I keep my score as borderline reject.

---

### Official Review · Reviewer_iovA · 2023-07-12

**Soundness:** 3 good
**Presentation:** 2 fair
**Contribution:** 4 excellent
**Rating:** 7
**Confidence:** 5

**Summary:**

The paper tackles the task of monocular depth estimation which is of fundamental importance in computer vision and has many downstream applications. Several recent works use bin-based approaches and follow the adaptive binning framework where the distribution of bin centers on the depth interval (that are treated as depth candidates) can vary per image or per-pixel.

This paper proposes a novel approach to the adaptive binning framework called IEBins. Instead of working with only one (potentially large) set of bins (per image or per pixel), IEBins refines the binning structure in an iterative manner. Starting with a coarse uniform division of the original depth interval (dmin, dmax), the idea is to recursively find and uniformly divide the ‘target’ bin by using the ’target’ bin as the new target depth interval for the next step. In addition, the target bin is made elastic (the new target depth interval’s ends are allowed to change) based on the uncertainty estimate.

The work achieves state-of-the-art results on popular benchmarks including KITTI and NYU-Depth-v2 and the authors promise to release the code and models publicly.

**Strengths:**

* State-of-the-art results. The work achieves SOTA in a highly competitive space of monocular depth estimation with over 7% improvement in RMSE over the prior SOTA. IEBins idea is also validated and shows about 2% improvement to adaptive bins baseline in fair settings.
* Interesting idea. The idea of recursive division of bins is obvious in retrospect yet creative. Elastic nature of bins based on uncertainty also makes sense and subtly introduces the ‘adaptive’ nature of bins.
* Well exploited ideas and good architecture design. Authors have well exploited the ideas from other works to obtain their goal. e.g. Iterative refinement using GRU, developed in separate field of optical flow estimation, has been introduced here in a well designed architecture.
* Potential to be foundational. Recently, depth estimation has seen foundational ideas like adaptive binning that serve as a starting point for a series of several follow up works. I believe the iterative refinement of bins can prove to be as influential.
* Good to see ablations on all bin types in fair settings.


**Weaknesses:**

* W1 **Incomplete literature review**. Authors have missed some important published works that are highly relevant to this work:
    * [a] - LocalBins (ECCV’22) also introduces the idea of “splitting” the bins and step-wise refinement of bins in a coarse-to-fine manner. IEBins and LocalBins are highly related and pursue the same goal but follow different approaches. Quantitative comparison with [a] and providing the insights to differences and similarities is highly suggested.
    * [b] - PixelFormer (WACV’23) is also based on the adaptive binning framework and introduces layer-wise refinement of “pixel queries”.
* W2 **Absence of quantitative evidence towards the working of the fundamental idea**. According to the proposed idea of iteratively making the ‘target’ depth interval smaller, the bin-width (or its median across an image) should exponentially decrease stage-by-stage, with some flexibility introduced by the elastic nature. At the same time, if the bins are too elastic i.e. final bin-widths are comparable to initial bin-widths (e.g. only 50% smaller than previous or so), then the iterative restrictive nature of depth search is invalidated. Although authors visualise the refinement process in Fig. 3 but these results can still be explained by ‘too elastic’ bins as the elastic-bin-adjustment varies spatially. Also, are the uncertainty 'heat maps' scaled individually or globally? If individually (which seems to be the case), then the uncertainty visualizations only show that uncertainty tends to take much higher values near edges, rather than an overall absolute decrease and therefore tell nothing about the bin widths. The authors are suggested to provide evidence to show that method actually works as described to avoid misleading conclusions. For example, via a simple line plot of evolution of target bin-width (multiple lines for pixels across a row or single line plot of median elastic bin width across image etc) through the course of refinement, or any other form that authors deem suitable that delivers the evidence clearly.
* W3 **Unsubstantiated claim at L44**. Authors claim that large number of bins are undesirable and use this as the main motivation for the work. Authors state at L43-44 that “Learning ambiguity grows rapidly as the number of bins increases” but fail to provide any explanation, reference or evidence. As this information can be potentially misleading, authors should either remove the claims or provide an explanation, reference or evidence towards the claim.

[a] Bhat, Shariq Farooq, Ibraheem Alhashim, and Peter Wonka. "Localbins: Improving depth estimation by learning local distributions." European Conference on Computer Vision. Cham: Springer Nature Switzerland, 2022.

[b] Agarwal, Ashutosh, and Chetan Arora. "Attention attention everywhere: Monocular depth prediction with skip attention." Proceedings of the IEEE/CVF Winter Conference on Applications of Computer Vision. 2023.

**Questions:**

* How do the bin-widths actually evolve during the iterative refinement? What is the mean of elasticity factor (new adjusted width divided by the bin-width at that stage with no elasticity)?
* What is the maximum bin-width in meters reached at the final stage e.g. for the NYU test set? This can also answer the elasticity to some extent.
* Why is a large number of bins undesirable? What is the evidence?
* How does IEBins compare with LocalBins?

**Limitations:**

The authors do not explicitly reflect upon the limitations or scope of the work. Refer to "weaknesses" for major concerns.

---

> ### Author Rebuttal · Authors · 2023-08-07
>
> ### __We thank our reviewer for the constructive feedback and comments.__
>
> ### _W1&Q4: Incomplete literature review. Comparison with LocalBins._
>
> A1: We will add these two interesting and relevant works to our revised version.
>
> Similarities with LocalBins: Both IEBins and LocalBins use a multi-stage fashion to refine the binning structure.
>
> Differences between them: At each stage, LocalBins divides all bins from the previous stage, and when the stage increases, the number of bins increases, while IEBins locates and divides the target bin only, and the number of bins is not changed at different stages. LocalBins refines the binning structure on multiple resolutions, while IEBins on the same resolution.
>
> Quantitative comparison with LocalBins: We follow the experimental settings in Table 5 and the results of Baseline + LocalBins on Abs Rel, RMSE, log10, $\delta  < 1.25$,  $\delta  < {1.25^2}$, $\delta  < {1.25^3}$ are 0.090, 0.319, 0.038, 0.932, 0.992, 0.998 on the NYU-Depth-v2 dataset, which are worse than those of Baseline + IEBins (0.087, 0.314, 0.038, 0.936, 0.992, and 0.998).
>
> ### _W2&Q1&Q2: Quantitative evidence towards the working of the fundamental idea, elasticity factor and maximum bin-width in meters reached at the final stage._
>
> A2: Nice comments. We randomly choose a sample from the NYU-Depth-v2 test set, and show the median elastic target bin width across the image for each stage, and corresponding uncertainty values and elasticity factors. We note that 9.9 is the original range size ($d_{max}$ 10 - $d_{min}$ 0.1) .
>
> |&emsp;&emsp;&emsp;&emsp;&emsp;&emsp;&emsp;&emsp; Stage1    &nbsp;  Stage2  &nbsp;   Stage3   &nbsp;  Stage4  &nbsp;   Stage5   &nbsp;   Stage6||
> |:---|:---|
> |Width (median)   &ensp;&ensp;&ensp; 9.9 &ensp;&ensp;&ensp; 2.561 &ensp;&ensp;&ensp; 0.756 &ensp;&ensp; 0.228 &ensp;&ensp; 0.073 &ensp;&ensp; 0.024|
> |Uncertainty (std)  &ensp;&ensp;&ensp; - &ensp;&ensp;&ensp;&ensp;&nbsp;  0.971 &ensp;&ensp;&nbsp; 0.281 &ensp;&ensp; 0.087 &ensp;&ensp; 0.029 &ensp;&ensp; 0.010|
> |Elasticity factor &ensp;&ensp;&ensp;&nbsp; &nbsp; - &ensp;&ensp;&ensp;&ensp; 4.139 &ensp;&ensp;&nbsp;&nbsp; 3.917 &ensp;&ensp; 4.222 &ensp;&ensp; 4.562 &ensp;&ensp;  4.800|
>
> It can be seen that as the stage increases, the elastic target bin widths and uncertainty values continue to decrease. The elasticity factors are between 3.9 and 4.8.
>
> The maximum bin widths of elastic target bin and newly divided bins for final stage are 0.046m and 0.0029m (0.046/16).
>
> ### _W3&Q3: Unsubstantiated claim at L44. Why is a large number of bins undesirable? What is the evidence?_
>
> A3: As presented in [1], the depth candidate corresponding to the peak point of the probabilistic distribution (or complementary cost distribution) may not be the true optimal depth candidate. However, the desired depth prediction can still be obtained after a linear combination of the probabilistic distribution and the depth candidates. In other words, there are many linear combinations for a set of depth candidates that can yield the desired depth prediction. When the number of bins increases, the combination between the probabilistic distribution and the depth candidates becomes more and more complex. Hence, we point out that ``this learning ambiguity grows rapidly as the number of bins increases''. Intuitively, it is much easier to classify the optimal candidate from a small set of depth candidates than from a large set of depth candidates. Hence, we choose small number of bins. To avoid potentially misleading, we will rephrase or remove this claim in the revision.
>
> [1]. Adaptive Unimodal Cost Volume Filtering for Deep Stereo Matching, AAAI2020.

---

> > ### Comment · Reviewer_iovA · 2023-08-16
> >
> > Thanks for your reply! Your response answered my queries.
> >
> > From the table you shared, it is evident that the idea works. Although the response to W3 makes sense, it is highly contextual and not an established fact e.g. AdaBins performs best at 256 bins (>> nbins in IEBins). So it is still suggested to rephrase or remove.
> >
> > LocalBins comes very close (equal in 3 out of 6 metrics) and makes the improvements weaker.
> >
> > The visualization can be better in all figures. The results in all cases should be normalized uniformly (e.g. use a dataset global min (e.g. 0.1 meters) and max depth (e.g. 10 meters) for depth maps, and not individual basis). This includes the uncertainty visualization as commented above.
> >
> > In summary, I believe the overall design of the proposed framework is quite nice and extendable. There are missing parts and weaknesses but the design and performance outweigh them. The paper is worth the acceptance.

---

> > > ### Author Response · Authors · 2023-08-16
> > >
> > > ### __Thank you very much for your feedback and encouraging words.__
> > >
> > > We will revise our paper accordlingly.

---

### Official Review · Reviewer_4WPj · 2023-07-14

**Soundness:** 2 fair
**Presentation:** 2 fair
**Contribution:** 2 fair
**Rating:** 4
**Confidence:** 4

**Summary:**

This paper proposes iterative elastic bins (IEBins), a multi-stage coarse-to-fine method for monocular depth estimation (MDE). It progressively searches the target depth bin on top of the previous step. To reduce the error accumulation in the iterations, an elastic bin is proposed whose width is adjusted based on the depth prediction uncertainty. Experiments on the KITTI, NYU-v2, and SUN RGB-D datasets show improved performances on MDE.

**Strengths:**

- The depth bin is adjusted elastically based on the depth prediction uncertainty.

**Weaknesses:**

- The contributions are not enough to my knowledge. I think only the elastic bins are new. The iterative manner for depth estimation by GRUs has been proposed in RAFT[28] and adopted in many follow-up works, e.g., RAFT-Stereo[29] and Itermvs[30]. The depth bins as shown in Eq 1,2,3&4 have been investigated in Adabins [3]. As for the framework of the feature extractor and GRU-based layers, standard implementations are involved. The multi-stage depth estimation pattern was also seen in previous works, e.g., CasMVS [Gu et al, cvpr2020] and IterMVS[30].
- Some minor ones:
  - line 141: a RGB --> an RGB
  - line 151: during iteration --> during iterations

**Questions:**

- Have other depth uncertainty presentations been investigated or compared with the variance of the probabilistic distribution used in Eq 6?

**Limitations:**

- N/A for the limitations.
- I suggest the authors provide the failure cases (and the corresponding explanations) of the proposed IEBins, w.r.t. for example, the number of bins and the elastic width of bins.

---

> ### Author Rebuttal · Authors · 2023-08-07
>
> ### __We thank our reviewer for the constructive feedback and comments.__
>
> ### _W1: The contributions are not enough to my knowledge._
>
> A1: Apologies for not stating the contributions clearly. In this work, we introduce a novel iterative elastic bins (IEBins) strategy for monocular depth estimation.  Instead of using only one set of bins like AdaBins [3], the IEBins refines binning structure in an iterative manner. The initialization stage makes a coarse uniform discretization of the full depth range and each subsequent stage iteratively locates and uniformly discretizes the target bin by using the target bin as the new depth range. While previous works CasMVS [Gu et al, cvpr2020] and IterMVS [30] also adopt a multi-stage fashion to estimate depth, they acquire a new depth range by centering on the current depth prediction and empirically setting the range size. In addition, the depth interval size of different stages at the same level in IterMVS [30] remains the same, while the bin width of our target bin is gradually reduced for a finer-grained depth search. To cope with the possible error accumulation during iterations, we further make the target bin elastic based on the depth uncertainty. To instantiate the IEBins, we design the network architecture by borrowing ideas from other fields such as optical flow estimation (RAFT [28]). Last but not least, the IEBins ranks 2nd and outperforms all previously published methods on the KITTI benchmark leaderboard at the submission time.
>
> ### _W2: Some minor ones: a RGB --> an RGB, during iteration --> during iterations._
>
> A2: We will fix these grammatical errors in the revision.
>
> ### _Q1: Have other depth uncertainty presentations been investigated or compared with the variance of the probabilistic distribution used in Eq 6?_
>
> A3: We have experimented with the predictive uncertainty [1] as well as using variance directly in our framework and find that using standard deviation of the probabilistic distribution works better.
>
> [1] . Depth Completion from Sparse LiDAR Data with Depth-Normal Constraints, CVPR 2019.
>
> ### _L1: I suggest the authors provide failure cases (and the corresponding explanations) of IEBins._
>
> A4: We will revise our paper according to this nice advice.

---

### Official Review · Reviewer_s85F · 2023-07-25

**Soundness:** 3 good
**Presentation:** 3 good
**Contribution:** 3 good
**Rating:** 5
**Confidence:** 4

**Summary:**

The paper proposes an iterative elastic bins (IEBins) strategy for monocular depth estimation. The IEBins use a small number of bins adaptively at each iteration. It use a GRU to predict the depth distribution at each stage. The authors conduct experiments on 3 commonly used datasets and the proposed method shows better results than previous methods.

**Strengths:**

The paper is well written and easy to follow.
The idea of using iterative update is interesting.
The experiments and ablation study are thorough and validate the proposed method's effectiveness.



**Weaknesses:**

1. The motivation of iterative updates is not very clear. RAFT used iterative updates because at each iteration, a new cost volume can be constructed based on current prediction. But I don't see where the new information comes from in MDE.
2. Based on 1. Maybe the improved performance come from the depth candidates projects since the model knows what the depth is. Would be interesting to see how the model performance by using a positional encoding of depth candidates to other non-iterative methods, such as adabins.

**Questions:**

L44. "This learning ambiguity grows rapidly as the number of bins increases, exacerbating the difficulty of model convergence and prone to overfitting." Why the ambiguity grows in this case? And why the proposed iterative method doesn't lead to increased ambiguity?



**Limitations:**

1. Currently only the depth maps are shown. It would be great to compare the point cloud with other methods since it's much easier to see the 3D structure in point cloud.
2. How is the improved depth estimation benefits in downstream task (compared to other MDE methods)? For example, could it be used for SLAM or 3D reconstruction from multiview images.

---

> ### Author Rebuttal · Authors · 2023-08-07
>
> ### __We thank our reviewer for the constructive feedback and comments.__
>
> ### _W1: The motivation of iterative updates is not very clear. I don't see where the new information comes from in MDE._
> A1: The IEBins embodies the idea of iterative division of bins. Each stage first divides the elastic target bin (the bin in which the current depth prediction is located) and then feeds the updated depth candidates into the iterative optimizer as new information for finer-grained depth search.
>
> ### _W2: Would be interesting to see how the model performance by using a positional encoding of depth candidates to other non-iterative methods, such as adabins._
> A2: Nice comment. We have verified the model performance by using a positional encoding of depth candidates to AdaBins while there is little change in performance.
>
> ### _Q1: Why the ambiguity grows in this case? And why the proposed method doesn't lead to increased ambiguity?_
> A3: As presented in [1], the depth candidate corresponding to the peak point of the probabilistic distribution (or complementary cost distribution) may not be the true optimal depth candidate. However, the desired depth prediction can still be obtained after a linear combination of the probabilistic distribution and the depth candidates. In other words, there are many linear combinations for a set of depth candidates that can yield the desired depth prediction. When the number of bins increases, the combination between the probabilistic distribution and the depth candidates becomes more and more complex. Hence, we point out that ``this learning ambiguity grows rapidly as the number of bins increases”. The proposed IEBins uses multiple small number of bins, instead of one large standard number of bins. To avoid potentially misleading, we will rephrase or remove this claim in the revision.
>
> [1]. Adaptive Unimodal Cost Volume Filtering for Deep Stereo Matching, AAAI 2020.
>
> ### _L1: It would be great to compare the point cloud with other methods._
>
> A4: We have shown point cloud comparison in Figs. 3 and 4 of supplementary material.
>
> ### _L2：How is the improved depth estimation benefits in downstream task (compared to other MDE methods)?_
>
> A5: We will add necessary discussion on our future works according to this nice advice in the revised version.

---

> > ### Comment · Reviewer_s85F · 2023-08-14
> >
> > Thanks for your reply. I think the MDE metrics are kind of saturated and would be great to show the benefits of improvements in downstream task such as slam or 3D reconstruction.

---

> > > ### Author Response · Authors · 2023-08-18
> > >
> > > ### __Thank you very much for your feedback.__
> > >
> > > ### _L2: I think the MDE metrics are kind of saturated and would be great to show the benefits of improvements in downstream task such as slam or 3D reconstruction._
> > >
> > > A1: As suggested,  we integrate IEBins and NeWCRFs [6] into ORB-SLAM2 [1] in the RGB-D setting and evaluate the visual odometry performance on the KITTI odometry dataset. We report results on keyframes (selected by the ORB-SLAM2) and on all frames of sequences 01-10. The ATE (m) metric is used. ''key'' and ''all'' stand for keyframes and all frames, respectively.
> > >
> > > |Seq &nbsp;&nbsp;&nbsp;&nbsp; IEBins (key) &nbsp; NeWCRFs (key) &nbsp;&nbsp;&nbsp;&nbsp;&nbsp;&nbsp; IEBins (all) &nbsp; NeWCRFs (all)||
> > > |:---|:---|
> > > |01 &nbsp;&ensp;&ensp;&ensp;&ensp; 117.06 &nbsp;&ensp;&ensp;&ensp;&ensp;&ensp;&ensp;&ensp;&ensp; 536.53 &nbsp;&ensp;&ensp;&ensp;&ensp;&ensp;&ensp;&ensp;&ensp;&ensp; 125.09 &nbsp;&ensp;&ensp;&ensp;&ensp;&ensp;&ensp; 583.20|
> > > |02 &nbsp;&ensp;&ensp;&ensp;&ensp; 12.22 &nbsp;&ensp;&ensp;&ensp;&ensp;&ensp;&ensp;&ensp;&ensp;&ensp; 13.32 &nbsp;&ensp;&ensp;&ensp;&ensp;&ensp;&ensp;&ensp;&ensp;&ensp;&ensp;&nbsp; 13.59 &nbsp;&ensp;&ensp;&ensp;&ensp;&ensp;&ensp;&ensp; 13.97|
> > > |03 &nbsp;&ensp;&ensp;&ensp;&ensp;&nbsp; 6.72 &nbsp;&ensp;&ensp;&ensp;&ensp;&ensp;&ensp;&ensp;&ensp;&ensp;&ensp;&nbsp; 8.31 &nbsp;&ensp;&ensp;&ensp;&ensp;&ensp;&ensp;&ensp;&ensp;&ensp;&ensp;&ensp;&nbsp; 7.15 &nbsp;&ensp;&ensp;&ensp;&ensp;&ensp;&ensp;&ensp;&ensp; 9.04|
> > > |04 &nbsp;&ensp;&ensp;&ensp;&ensp; 16.70 &nbsp;&ensp;&ensp;&ensp;&ensp;&ensp;&ensp;&ensp;&ensp;&ensp; 31.56 &nbsp;&ensp;&ensp;&ensp;&ensp;&ensp;&ensp;&ensp;&ensp;&ensp;&ensp;&nbsp; 16.61 &nbsp;&ensp;&ensp;&ensp;&ensp;&ensp;&ensp;&ensp; 30.59|
> > > |05 &nbsp;&ensp;&ensp;&ensp;&ensp;&nbsp; 8.10 &nbsp;&ensp;&ensp;&ensp;&ensp;&ensp;&ensp;&ensp;&ensp;&ensp;&ensp;&nbsp; 8.05 &nbsp;&ensp;&ensp;&ensp;&ensp;&ensp;&ensp;&ensp;&ensp;&ensp;&ensp;&ensp;&nbsp; 7.56 &nbsp;&ensp;&ensp;&ensp;&ensp;&ensp;&ensp;&ensp;&ensp; 7.86|
> > > |06 &nbsp;&ensp;&ensp;&ensp;&ensp;&nbsp; 1.32 &nbsp;&ensp;&ensp;&ensp;&ensp;&ensp;&ensp;&ensp;&ensp;&ensp;&ensp;&nbsp; 0.96 &nbsp;&ensp;&ensp;&ensp;&ensp;&ensp;&ensp;&ensp;&ensp;&ensp;&ensp;&ensp;&nbsp; 1.35 &nbsp;&ensp;&ensp;&ensp;&ensp;&ensp;&ensp;&ensp;&ensp; 0.95|
> > > |07 &nbsp;&ensp;&ensp;&ensp;&ensp;&nbsp; 2.48 &nbsp;&ensp;&ensp;&ensp;&ensp;&ensp;&ensp;&ensp;&ensp;&ensp;&ensp;&nbsp; 3.09 &nbsp;&ensp;&ensp;&ensp;&ensp;&ensp;&ensp;&ensp;&ensp;&ensp;&ensp;&ensp;&nbsp; 2.55 &nbsp;&ensp;&ensp;&ensp;&ensp;&ensp;&ensp;&ensp;&ensp; 3.24|
> > > |08 &nbsp;&ensp;&ensp;&ensp;&ensp; 10.89 &nbsp;&ensp;&ensp;&ensp;&ensp;&ensp;&ensp;&ensp;&ensp;&ensp;&nbsp;&nbsp; 9.82 &nbsp;&ensp;&ensp;&ensp;&ensp;&ensp;&ensp;&ensp;&ensp;&ensp;&ensp;&nbsp;&nbsp; 11.06 &nbsp;&ensp;&ensp;&ensp;&ensp;&ensp;&ensp;&ensp; 9.90|
> > > |09 &nbsp;&ensp;&ensp;&ensp;&ensp;&nbsp; 5.44 &nbsp;&ensp;&ensp;&ensp;&ensp;&ensp;&ensp;&ensp;&ensp;&ensp;&ensp;&nbsp; 7.61 &nbsp;&ensp;&ensp;&ensp;&ensp;&ensp;&ensp;&ensp;&ensp;&ensp;&ensp;&ensp;&nbsp; 5.68 &nbsp;&ensp;&ensp;&ensp;&ensp;&ensp;&ensp;&ensp;&ensp; 7.67|
> > > |10 &nbsp;&ensp;&ensp;&ensp;&ensp;&nbsp; 7.21 &nbsp;&ensp;&ensp;&ensp;&ensp;&ensp;&ensp;&ensp;&ensp;&ensp;&ensp; 11.73 &nbsp;&ensp;&ensp;&ensp;&ensp;&ensp;&ensp;&ensp;&ensp;&ensp;&ensp;&ensp; 8.24 &nbsp;&ensp;&ensp;&ensp;&ensp;&ensp;&ensp;&ensp; 12.66|
> > >
> > > As we can see, our IEBins either significantly exceeds the NeWCRFs or achieves on par performance with the latter.
> > >
> > > [1] ORB-SLAM2: an Open-Source SLAM System for Monocular, Stereo and RGB-D Cameras, IEEE Transactions on Robotics, 2017
> > >
> > > We further evaluate IEBins and NeWCRFs [6] on the single-view reconstruction task. The RMSE (m) metric is used.
> > >
> > > |Method &nbsp;&nbsp; RMSE (NYU) &nbsp;&nbsp; RMSE (KITTI) &nbsp; ||
> > > |:---|:---|
> > > |IEBins &nbsp;&ensp;&ensp;&ensp;&ensp; 0.195 &nbsp;&ensp;&ensp;&ensp;&ensp;&ensp;&ensp;&ensp;&ensp; 1.481|
> > > |NeWCRFs &nbsp;&nbsp;&nbsp; 0.205 &nbsp;&ensp;&ensp;&ensp;&ensp;&ensp;&ensp;&ensp;&ensp;  1.526|
> > >
> > > As we can see, our IEBins exceeds NeWCRFs by 4.9% and 2.9% on NYU and KITTI datasets, respectively.

---

### Decision · Program_Chairs · 2023-09-21

**Decision:**

Accept (poster)

**Comment:**

This paper received 4x positive ratings (2x borderline accepts, 1x accept and 1x strong accept) and 2x negative ratings (2x borderline rejects). The reviewers who gave the positive comments mention that: The idea of using iterative update is interesting, the idea of recursive division of bins is obvious in retrospect yet creative. The method outperforms prior work across multiple standard evaluation settings; The method is interpretable, producing confidence for different regions and steadily improving depth predictions; The introduced method is very fast; Good results and good ablation study on all bin types in fair settings; Well exploited ideas and good architecture design. The negative comments are: Elastic bins are new. The iterative manner for depth estimation by GRUs has been proposed in RAFT[28] and adopted in many follow-up works. Not totally convinced by additional contributions (such as GRU-based optimizer) and analysis. One of the positive reviewer also mentioned that the idea is not totally new, but it's from other areas, e.g. optical flow, sfm, etc. Considering that the positive comments outweigh the negative comments in the reviews, the AC decide to follow the majority vote to accept the paper.